



# Mapping soil organic carbon fractions for Australia, their stocks and uncertainty

Mercedes Román Dobarco[1], Alexandre M.J-C. Wadoux[1], Brendan Malone[2], Budiman Minasny[1], Alex B. McBratney[1], Ross Searle[3]

[1]Sydney Institute of Agriculture & School of Life and Environmental Sciences, The University of Sydney, 1 Central Avenue, Eveleigh, 2015, NSW, Australia

[2]CSIRO Agriculture and Food, Black Mountain, ACT, Australia

[3]CSIRO Agriculture and Food, 306 Carmody Road, St Lucia, QLD, Australia.

∗*Correspondance to:* Mercedes Román Dobarco (mercedes.romandobarco@sydney.edu.au)

**Abstract.** Soil organic carbon (SOC) is the largest terrestrial carbon pool. SOC is composed of a continuum set of compounds with different chemical composition, origin and susceptibilities to decomposition, that are commonly separated into pools characterised by different responses to anthropogenic and environmental disturbance. Here we map the contribution of three SOC fractions to the total SOC content of Australia's soils. The three SOC fractions: mineral-associated organic carbon (MAOC), particulate organic carbon (POC) and pyrogenic organic carbon (PyOC), represent SOC composition with distinct turnover rates, chemistry, and pathway formation. Data for MAOC, POC, and PyOC were obtained with near- and mid-infrared spectral models calibrated with measured SOC fractions. We transformed the data using an isometric log-ratio transformation (ilr) to account for the closed compositional nature of SOC fractions. The resulting, back-transformed ilr components were mapped across Australia. SOC fraction stocks for the 0-30 cm were derived with maps of total organic carbon concentration, bulk density, coarse fragments and soil thickness. Mapping was done by quantile regression forest fitted with the ilr transformed data and a large set of environmental variables as predictors. The resulting maps along with the quantified uncertainty show the unique spatial pattern of SOC fractions in Australia. MAOC dominated the total SOC with an average of 59% ±17.5%, whereas 28% ± 17.5% was PyOC and 13% ± 11.1% was POC. The allocation of TOC into the MAOC fractions increased with depth. SOC vulnerability (i.e., POC/[MAOC + PyOC]) was greater in areas with Mediterranean and temperate climate. TOC and the distribution among fractions were the most influential variables on SOC fraction uncertainty. Further, the diversity of climatic and pedological conditions suggests that different mechanisms will control SOC stabilisation and dynamics across the continent, as shown by the model covariates importance metric. We estimated the total SOC stocks (0-30 cm) to be 12.7 Pg MAOC, 2 Pg POC and 5.1 Pg PyOC, which is consistent with previous estimates. The maps of SOC fractions and their stocks can be used for modelling SOC dynamics and forecasting changes in SOC stocks as response to land use change, management, and climate change.



Keywords: Soil organic carbon, SOC fractions, particulate organic carbon, mineral-associated organic carbon, pyrogenic organic carbon, digital soil mapping.

# 1 Introduction

Soils are the main organic carbon pool in terrestrial ecosystems, storing around two thirds of the total C. Global soil organic carbon (SOC) stock is estimated to be around 1500 PgC for the first metre of soil (Jobbagy and Jackson, 2000), with other estimates ranging from 504 to 3000 PgC (Scharlemann et al., 2014). Changes in SOC storage and dynamics can alter the ecosystem C balance and determine whether soils become C sources or sinks from local to global scale (Friedlingstein et al., 2020). SOC is strongly linked to most soil properties and functions (e.g., nutrient and water storage and cycling, habitat provisioning and biodiversity) (Van Leeuwen et al., 2019), and hence is used as a general indicator of soil quality/capacity (Schoenholtz et al., 2000; Bunemann et al., 2018) and soil health/condition.

SOC is the main component of soil organic matter, which is a continuum of compounds with different chemical compositions, origin (aboveground litter, dead roots, rhizodeposition, microbial-derived), degree of microbial processing and decomposition, and turnover times (Lehmann and Kleber, 2015). SOC is protected against microbial decomposition by several stabilisation mechanisms which have been generally grouped into: 1) selective preservation due to biochemical recalcitrance, 2) chemical stabilisation via interactions between organic compounds and mineral surfaces or metal cations, and 3) physical protection by the inaccessibility of decomposers to the organic matter (Sollins et al., 1996; Von Lutzow et al., 2006; Rowley et al., 2018). Spatial inaccessibility and interactions between the mineral surfaces of silt and clay-sized particles and organic compounds are considered the major mechanisms for mid and long-term SOC stability (Von Lutzow et al., 2006), whereas an increasing body of literature questions the relevance of biochemical recalcitrance for long-term persistence of SOC (Schmidt et al., 2011).

A myriad of physical, chemical, or combined fractionation methods have been designed for separating SOC into operational pools characterized by specific stabilization mechanisms, chemical composition, and distinct turnover rates, and yet it is difficult to isolate fractions that correspond to functional SOC pools (Lutzow et al., 2007). Some fractionation schemes were adapted to quantify conceptual SOC pools from established C dynamics models, e.g., the Rothamsted C model (RothC, Jenkinson and Rayner (1977)) from measured SOC fractions (Poeplau et al., 2013; Zimmermann et al., 2007). However, there can be some discrepancies between the predicted SOC pools when the model is initialized with modelled SOC pools from equilibrium runs or from measured SOC fractions (Poeplau et al., 2013). Other biogeochemical models have been conceptualized and calibrated with functional and measurable SOC fractions to overcome these differences (Robertson et al., 2019), but often require the determination of many SOC fractions. A comparison of several fractionation schemes (Poeplau et al., 2018) suggests that size separation into silt + clay (<53 μm) (i.e., mineral-associated organic carbon (MAOC)), and sand-sized particles (>53 μm) (i.e., particulate organic carbon (POC)), may suffice to differentiate pools with distinct turnover rates, chemistry, and pathway formation (Lavallee et al., 2020). MAOC is predominantly composed of low molecular weight molecules of microbial origin (e.g., microbial metabolites, necromass) (Miltner et al., 2012; Kallenbach et al., 2016; Liang et



al., 2019), leachates from plant litter, and rhizodeposition (Villarino et al., 2021), which are protected through sorption to mineral surfaces or occluded inside microaggregates (Lavallee et al., 2020). POC is mainly composed of partially decomposed

plant litter and roots and fungal-derived compounds (Baisden et al., 2002; Gregorich et al., 2006; Geng et al., 2019; Villarino et al., 2021), that can be found free or inside macroaggregates (Rabbi et al., 2014). MAOC has a lower C:N ratio, a higher proportion of microbial-derived and proteinaceous compounds (Kleber et al., 2007; Knicker, 2011) and a longer mean residence time (decades to centuries) than POC (mean residence time of years to decades) (Baisden et al., 2002; Gregorich et al., 2006; Von Lützow et al., 2007; Heckman et al., 2022). Separating SOC into POC and MAOC has been proposed as a

simple and effective way to conceptualize and model SOC dynamics (Lavallee et al., 2020), and has been applied to predict the vulnerability of SOC to future climate scenarios (Lugato et al., 2022).

In Australia, the long history of burning suggests pyrogenic organic carbon (PyOC) as an important component (Lehmann et al., 2008). PyOC refers to charred residues derived from incomplete combustion of organic matter (also known as charcoal or black carbon) (Lutfalla et al., 2017). PyOC can be found in both POC and MAOC fractions (Lavallee et al., 2019). PyOC is

comprised by a continuum of organic compounds thermally altered by fire, and its chemical composition and pool size depends on the technique used for its determination(Zimmerman and Mitra, 2017). PyOC is considered a relatively stable SOC fraction with a turnover time that ranges from decades to centuries (Singh et al., 2012), protected from decomposition by the biochemical recalcitrance of condensed aromatic C (Lavallee et al., 2019). However, turnover rates previously assessed from centuries to multi-millenia may have overestimated its persistence in soil (Singh et al., 2012; Lutfalla et al., 2017). PyOC

represents on average 14-26% of total SOC  and can reach up to 60-80% of SOC (Lehmann et al., 2008; Reisser et al., 2016). Globally, PyOC stocks are not controlled by fire intensity and return interval, but are explained by soil properties and climate (Reisser et al., 2016). However, in systems with local records of fire history PyOC content increased in sites with high-frequency fires (Reisser et al., 2016). In Australia, fire is an important driver of ecological processes, and bush-fire frequency has increased in recent decades (Dutta et al., 2016). Hence, PyOC can represent a large proportion of total SOC in some

Australian regions (Lehmann et al., 2008).

Mapping and quantifying the stocks of SOC fractions can be used for modelling SOC dynamics and forecasting changes in SOC stocks as a response to land-use change, management, and climate change (Lugato et al., 2022; Xu et al., 2011; Wiesmeier et al., 2016). High-resolution maps of SOC fractions can inform agricultural management at the farm scale and be incorporated in the design of climate change and soil security policies at the state and national level. The objective of this study was to map

the contribution of SOC fractions (MAOC, POC, and PyOC) to the total SOC in the top 30 cm and update the Soil and Landscape Grid of Australia (SLGA) (Grundy et al., 2015) products for SOC fraction stocks (Viscarra Rossel et al., 2019). These digital soil maps will be part of the v2.SLGA for Australia, following the principles of transparency, reproducibility, and updatability as new data become available (Malone and Searle, 2021).



## 2 Materials and Methods

The methodology implemented in this study consists of two main steps: 1) prediction of SOC fractions across the soil spectral libraries available for Australia with different spectral models calibrated with measured SOC fraction data from the Australian Soil Carbon Research Program (SCaRP) (Baldock et al., 2013a), and 2) mapping of the proportions of SOC fractions after applying the isometric log-ratio transformation (ilr) to account for the compositional nature of the data (i.e., the proportion of SOC fractions sum to 100%). Previous studies modelled SOC fractions stocks or concentrations (Sanderman et al., 2021;

Viscarra Rossel et al., 2019), but we found some difficulties in implementing such an approach (see Sect. 2.5) and hence mapped proportions of SOC. Finally, we calculated SOC fraction stocks for the 0-30 cm using data from SLGA maps of total organic carbon (TOC) concentration (Wadoux et al., 2022), bulk density (Viscarra Rossel et al., 2015), coarse fragments (this study) and soil thickness (Malone and Searle, 2020).

### 2.1 Study area

The study area covers the continent of Australia, including Tasmania and near-shore small islands. In Australia there are six major climatic regions following the Köppen classification. In the north, there are hot humid summers in equatorial, tropical and subtropical regions. Summers are hot and dry, with mild to cold winters in grasslands and desert regions in the interior. Temperate areas in the south coastal band have cold winters and warm summers, that are mild at higher elevations and latitudes. Vast areas of the continent are very dry, with precipitation increasing towards the coast and elevation in the mountains of the

eastern uplands. Australia is characterised by flat and low relief vast areas where the tectonic stability and lack of glaciation have preserved a deeply weathered mantle that dates from the Tertiary (Mckenzie et al., 2004). The distribution of soils in many regions depends on the stripping of this weathered mantle, while in other areas the dominant drivers of landforms and soils are water, fluvial, and aeolian erosion and depositional processes (McKenzie et al., 2004). Younger soils are found mainly in the east, along the Great Divide (McKenzie et al., 2004). The dominant soil orders according to the Australian Soil

Classification (Isbell et al., 1997) are Kandosol (30 % of the area), Tenosol (20 %), Vertosol (14 %), Rudosol (8 %), Calcarosol (8 %) and Chromosol (7 %) (Searle, 2021). In 2015-2016 most of the area was dedicated to primary production, with 48 % of the continent used for livestock grazing (42 % on native vegetation and 6 % modified pastures), less than 5 % dedicated to dryland and irrigated cropping. About 9.5% of Australia is allocated to nature conservation and 18 % are protected managed resources (ABARES, 2022).

**2.2 SCaRP dataset – sampling design and SOC fractionation scheme**

The soil sampling design and SOC fractionation protocol of SCaRP are described in detail in Sanderman et al. (2011) and Baldock et al. (2013c). The objective of SCaRP was to characterize SOC stocks and composition of agricultural topsoils (0-30 cm) and their response to agricultural practices. Plots of 25 m$^2$ were laid in 4526 sites representative of the different combinations of agricultural management and soil types across Australia. At each plot, soil samples were collected randomly





at 10 nodes of a 5 m x 5 m grid with a soil corer (≥ 4 cm diameter) by depth interval (0-10, 10-20, 20-30 cm) and composited
      by layer. In addition, at least three measurements of bulk density by depth interval were taken at each plot, although the
      methods varied between states to accommodate the soil type. A subset of 312 samples representative of the range of TOC
      content found among the whole dataset (N = 24,495 samples) were subject to fractionation  (Baldock et al., 2013c). The size
      and chemical fractionation scheme divided SOC into MAOC, POC and PyOC. MAOC and PyOC were originally defined as

humic organic carbon and resistant organic carbon in Baldock et al. (2013c) but we changed the terminology to match the
      recent literature. A 10-g aliquot of air-dried soil ≤ 2 mm was dispersed with 5 g L$^{-1}$ sodium hexametaphosphate and separated
      into coarse (>50 μm) and fine (<50 μm) fractions with wet sieving using an automated sieve shaker system (Baldock et al.,
      2013b). The TOC concentrations of the coarse and fine fractions were analysed with high-temperature oxidative combustion
      after the removal of inorganic carbon with 5-6 % H$_2$SO$_3$ if carbonates were present (method 6B3a, Rayment and Lyons (2011)).

The content of poli-aryl C was determined with solid-state $^{13}$C NMR (nuclear magnetic resonance spectroscopy) and used as
      an estimate of PyOC. POC and MAOC contents (mg C-fraction g$^{-1}$ soil) were calculated by subtracting the proportion of PyOC
      in each fraction (Baldock et al., 2013c).

### 2.3 Spectral datasets and harmonization

      Four spectral soil datasets were combined in this study: SCaRP, the Australian Soil Archive mid-infrared (MIR) spectral library
(AusSpecMIR) (Hicks et al., 2015), an additional MIR library of specimens from the Australian Soil Archive that were not
      included in the Hicks et al. (2015) work (AusSpecMIR2) and the Australian Soil Archive visible and near-infrared (vis-NIR)
      spectral library (AusSpecNIR) (Viscarra Rossel and Hicks, 2015).

      Multiple soil samples (approximately 700) were represented both in AusSpecMIR and AusSpecNIR, and duplicates were
      removed from the AusSpecNIR for the subsequent modelling. The AusSpecNIR library includes 433 samples from SCaRP.
The soil samples come from different soil surveys and projects carried out by Australian states and federal agencies. Notably,
      the AusSpecNIR includes samples from the Terrestrial Ecosystem Research Network (TERN) Ecosystem Surveillance plots
      (Sparrow et al., 2020; Malone et al., 2020), increasing the representation of non-agricultural soils (rangelands, forests) in the
      calibration data (Figure 1). Vegetation and soils are sampled in more than 800 permanent plots laid with stratified random
      sampling across bioregions (i.e., zones with similar landform, vegetation, and climate, analogous to ecoregion) (Sparrow et
al., 2020). Soil samples are taken from three depth intervals (0-10, 10-20, 20-30 cm) in nine locations within a 100 m x 100 m
      plot (Sparrow et al., 2020), for a total of 15,157 soil samples at 5,711 georeferenced sampling points.





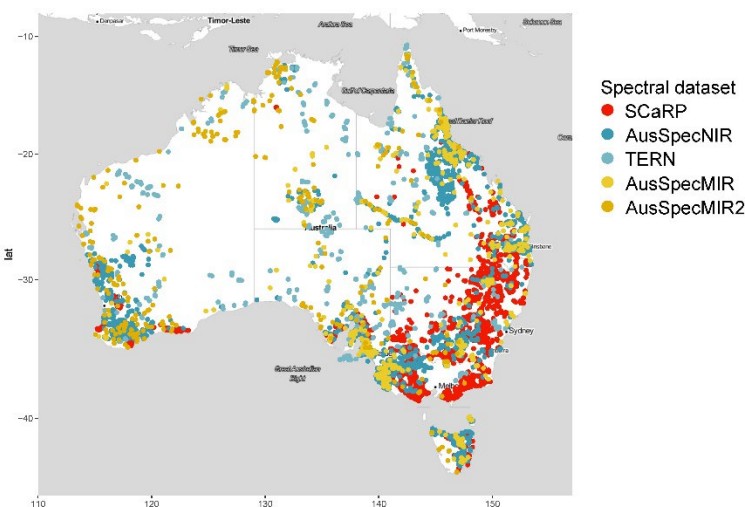

**Figure 1: Soil spectral datasets combined for mapping soil organic carbon fractions. TERN samples are part of the AusSpecNIR**
**library.**

The SCaRP spectral library was produced with a Nicolet 6700 FTIR spectrometer (Thermo Fisher Scientific Inc., Waltham, MA, USA) equipped with a KBr beam-splitter, a DTGS detector and an AutoDiff Automated diffuse reflectance accessory (Pike Technologies, Madison, WI, USA). Spectra were acquired over the range 8000-400 cm$^{-1}$ with a resolution of 8 cm$^{-1}$. The

spectrum of each soil sample was the average of 60 scans. The AusSpecMIR spectral library was created with a Bruker FT-IR Vertex70 spectrometer. This instrument is fitted with a mercury cadmium telluride detector that is liquid $N_2$ cooled to improve signal-to-noise ratio and has a spectral range of 7500-600 cm$^{-1}$ at 4 cm$^{-1}$ resolution. AusSpecMIR2 were measured with a Bruker FT-IR Invenio spectrometer with a similar liquid $N_2$ cooling system and spectral range of 7500-600 cm$^{-1}$ at 4 cm$^{-1}$ resolution. Soil samples from AusSpecMIR and AusSpecMIR2 were scanned per quadruplicate. Diffuse reflectance spectra

for the AusSpecNIR library were collected with a Labspec® vis–NIR spectrometer (PANalytical Inc., Boulder, CO, USA) for the range 350–2500 nm at 1 nm resolution. Each spectrum was the average of 30 scans and 4 spectra were collected for each soil sample (Viscarra Rossel and Hicks, 2015). To apply the SOC fraction predictive models calibrated with SCaRP data to the spectral datasets measured with a different instrument, we harmonised the spectra from SCaRP, AusSpecMIR and AusSpecMIR2 with piecewise direct standardization (PDS) (Bouveresse and Massart, 1996; Ge et al., 2011). Approximately

200 soil samples from the SCaRP dataset were measured with the Bruker FT-IR Vertex70 spectrometer, following the estimate of parameters to harmonise the SCaRP library with AusSpecMIR. Similarly, the AusSpecMIR2 was harmonised with AusSpecMIR with PDS using a standard of 300 soil samples measured with both instruments (Table 1).



## 2.4 Spectral models and predictions of SOC fractions

SOC fractions contents (mg C g$^{-1}$ soil) of SCaRP samples were estimated with partial least squares regression (PLSR) models
developed by Baldock et al. (2013b). Baldock et al. (2013b) reported cross-validation results (squared-root transformed SOC
fraction contents) with R$^2$ values of 0.84, 0.88 and 0.85, and RMSE of 0.43, 0.40 and 0.32 for POC, MAOC and PyOC
respectively.

SOC fractions content for AusSpecMIR and AusSpecMIR2 were estimated with a new set of spectral predictive functions.
These functions were developed with 200 SCaRP samples with data on SOC fractions concentration (mg C-SOC fraction g$^{-1}$
soil), total organic carbon (TOC) concentration (mg C g$^{-1}$ soil), and harmonised spectra. The contents of SOC fractions were
converted into percentages of TOC (summing up to 100 %) and modelled as compositional data. Hence, the ilr transformation
was applied to the proportions of SOC fractions. The ilr generates a D-1 dimensional Euclidean vector, where D is the number
of variables in the original variables (Aitchison, 1986). PLSR models with bootstrapping were used to predict the ilr-
transformed SOC fractions. Validation statistics were calculated from the average of 50 model realisations on the back-
transformed data. The average RMSE for POC, MAOC and PyOC were 6.2, 7.7, and 6.4 %, respectively. The Lin's
concordance correlation coefficient ($\rho_c$) values were 0.86, 0.76 and 0.70, and R$^2$ of 0.74, 0.61, and 0.52. When the percentages
were back-transformed into SOC fraction contents (mg C-SOC fraction g$^{-1}$ soil), the RMSE were 2.1, 2.7, and 1.8 mg C g$^{-1}$
soil, $\rho_c$ of 0.92, 0.95, and 0.90, and R$^2$ of 0.85, 0.88, and 0.77 for POC, MAOC and PyOC respectively.

SOC fraction contents for the AusSpecNIR dataset were predicted with PLSR models calibrated with data on TOC
concentration and SOC fractions contents from 309 SCaRP samples. Before modelling, the spectra were trimmed to the 453-
2500 nm range, processed with a Savitsky-Golay smoothing filter and transformed from reflectance to absorbance. We also
applied the standard normal variate transformation and then wavelet coefficients were derived. Similarly, as for the MIR
datasets, spectral predictive functions were developed for ilr-transformed SOC fraction compositional data, and validation
statistics were calculated for the back-transformed data. The validation statistics for POC, MAOC and PyOC models were
respectively RMSE of 5.0, 6.4, and 5.8 %, $\rho_c$ of 0.92, 0.84, and 0.76, and R$^2$ of 0.83, 0.71, and 0.61 when validated as
proportions of TOC. When the validation statistics were calculated for SOC fraction contents, the  RMSE were 3.2, 4.0 and
2.3 mg C g$^{-1}$ soil and $\rho_c$ of 0.81, 0.87, 0.80, and R$^2$ of 0.65, 0.75 and 0.64 for POC, MAOC and PyOC.

**Table 1: Spectral datasets, harmonization and spectral predictive models used to map soil organic carbon fractions.**

| Dataset | Number of Sites | Number of soil samples | Spectra | Spectra harmonization | Spectral model | Reference |
|---|---|---|---|---|---|---|
| SCaRP | 4498 | 14426 | MIR | From SCaRP to AusSpecMIR with 200 | PLSR developed with 312 samples | Baldock et al. (2013b) |



| | | | | | | |
|---|---|---|---|---|---|---|
| | | | | samples scanned with both spectrometers | | |
| AusSpecMIR | 3976 | 719 | MIR | Standard | PLSR developed with 200 samples from SCaRP dataset. Models for ilr-transformed variables. | Malone and Wadoux (2021) |
| AusSpecMIR2 | 337 | 300 | MIR | From AusSpecMIR2 to AusSpecMIR with 300 samples scanned with both spectrometers | | |
| AusSpecNIR | 9,289 | 22,684 | Vis-NIR | | PLSR developed with 309 samples from SCaRP dataset. Models for ilr-transformed variables. | Malone (2021) |


## 2.5 SOC fractions data processing and depth standardisation

The georeferenced SOC fraction data from all libraries was projected to WGS 84 (EPSG:4326) and collated. The data were filtered and processed to harmonize units, duplicates and potentially wrong data entries (e.g., missing upper or lower horizon depths) were excluded. SOC predictions of the quadruplicated scans per soil sample were averaged, as well as multiple soil samples per horizon by sampling location. SOC fraction concentrations (mg C-SOC fraction $g^{-1}$ soil) were transformed to proportion of total SOC (% SOC). SOC vulnerability (Vp) was defined as the ratio of POC to the sum of MAOC and PyOC (Baldock et al., 2018), i.e., the proportion of the less protected SOC fraction (POC) to the SOC fractions with stability by physico-chemical protection and recalcitrance, and longer turnover rates. SOC fractions and Vp data were standardized to the first three depth intervals of the GlobalSoilMap specifications (0-5 cm, 5-15 cm, 15-30 cm) (Arrouays et al., 2014) with equal-area quadratic splines (Bishop et al., 1999) (Figure 2). For sampling locations with just one sampled horizon, this was converted to the GlobalSoilMap depth intervals (assigned to a single or several depth intervals) via aggregation with the *slab* function of the *aqp* R package (Beaudette et al., 2022). We calculated the mean and standard deviation of SOC fractions and Vp by GlobalSoilMap depth interval and by biome (Olson et al., 2001).





Figure 2: Calibration data standardized for the depths 0-5 cm, 5-15 cm, and 15-30 cm. Contribution of SOC fractions to total organic carbon (% TOC): mineral-associated SOC (MAOC), particulate organic carbon (POC), pyrogenic organic carbon (PyOC) and soil organic carbon vulnerability (Vp).



Modelling SOC fraction concentrations directly was the preferred option tested in preliminary work. We compared the sum of
the predicted SOC fraction contents with measured TOC. The Pearson's $r$ correlation coefficient was 0.56, but the sum of SOC
fractions showed some extreme values (Figure S1). We adjusted the SOC fraction contents with measured TOC data extracted
from the Soil Data Federator (Searle et al., 2021) and the Biome of Australian Soil Environments (BASE) contextual data
(Bissett et al., 2016). However, the resulting dataset had a reduced spatial coverage and the maps exhibited unrealistic patterns.
Hence, we preferred to use all available data and map the proportion of SOC fractions and SOC vulnerability.

The values of SOC proportions were modelled considering their compositional nature (i.e., multivariate data with positive
values that sum up to a constant, in the case of SOC fractions 100%). The set of all compositional observations ($S^D$) is a
simplex sample space, a subset of the real space $\mathbb{R}^D$. The ilr-transformation (Egozcue et al., 2003) transposes $S^D$ into
multidimensional real space ($\mathbb{R}^{D-1}$), without the collinearity problems associated with other transformations, such as the
centered log-ratio transformation (clr). Ilr is based on the choice of an orthonormal basis on the hyperplane in $\mathbb{R}^D$ formed by
the clr transformation. The ilr-transformation equation is defined:

$$z_i = ilr(x_i) = \sqrt{\frac{i}{i+1}} \ln\left(\frac{\sqrt[i]{\prod_{j=1}^{i} x_j}}{x_{i+1}}\right), i = 1, 2, \dots, D-1 \tag{1}$$

And the inverse equation is defined as follows (Filzmoser and Hron, 2008):

$$x_i = \frac{\exp(y_i)}{\sum_{i=1}^{D} \exp(y_i)} \; i = 1, 2, \dots, D \tag{2}$$

$$y_i = \sum_{j=i}^{D} \frac{z_j}{\sqrt{j\,(j+1)}} - \sqrt{\frac{i-1}{i}} z_{i=1} \; with \; z_0 = z_D = 0 \; for \; i = 1, 2, \dots, D \tag{3}$$

We applied the ilr-transformation with the *ilr* function of the *compositions* R package (Van Den Boogaart et al., 2022),
generating two variables hereafter referred to as ilr1 and ilr2. The predictions (ilr1 and ilr2) were back-transformed into
MAOC, POC, PyOC with the *ilrInv* function.

### 2.6 Coarse fragments

Data on the abundance of coarse fragments (particles > 2 mm) and gravimetric content (% weight) were extracted using the
TERN Soil Data Federator (https://esoil.io/TERNLandscapes/Public/Pages/SoilDataFederator/SoilDataFederator.html)
managed by CSIRO (Searle et al., 2021). The Soil Data Federator is a web API that compiles soil data from different institutions
and government agencies throughout Australia. The abundance (% volume) is assessed visually in the field as part of the soil
profile description using standards described in the Australian Soil and Land Survey field Handbook (National Committee on
Soil and Terrain, 2009). The abundance of rock fragments per soil horizon on the cut surface of the soil profile was grouped
into six categories: very few (0-2 %), few (2-10 %), common (2-20 %), many (20-50 %), abundant (50-90 %) and very
abundant (> 90%). The gravimetric content (% mass) is measured in the laboratory as percent mass of coarse fragments
(particles > 2 mm) from the whole soil. Here, we take the profile surface abundance of coarse fragments as a proxy for





volumetric coarse fragments (CF$_{Vol}$). The data was cleaned and processed to exclude duplicates and wrong data entries (e.g., missing values). The observations of CF$_{Vol}$ (%) were converted into GlobalSoilMap depth intervals with the *slab* function of

the *aqp* R package (Beaudette et al., 2022), assigning the most probable class to each depth interval. The gravimetric coarse fragments were also standardized to the GlobalSoilMap depth intervals with equal-area quadratic splines (Bishop et al., 1999). Observations of gravimetric coarse fragment content ($CF_{Mass}$) were transformed into volumetric with Eq.(4):

$$CF_{Vol}\ (\%) = \frac{Vol_{CF}}{Vol_{WhSoil}}\frac{Mass_{CF}\,/\,\rho_{CF}}{Mass_{WhSoil}\,/\,\rho_{WhSoil}} = \frac{CF_{Mass} \times \rho_{WhSoil}}{\rho_{CF}} \qquad (4)$$

where $\rho_{WhSoil}$ is the bulk density prediction for bulk soil from SLGA (Viscarra Rossel et al., 2015), $\rho_{CF}$ is assumed to be 2.65

g cm$^{-3}$ (Hurlbut and Klein (1977) in Mckenzie et al. (2002)) and $CF_{Vol}$ is the volumetric coarse fragment content (continuous),which was assigned to the corresponding class. This resulted in CF$_{Vol}$ observations for 110,308 locations (Figure 3).

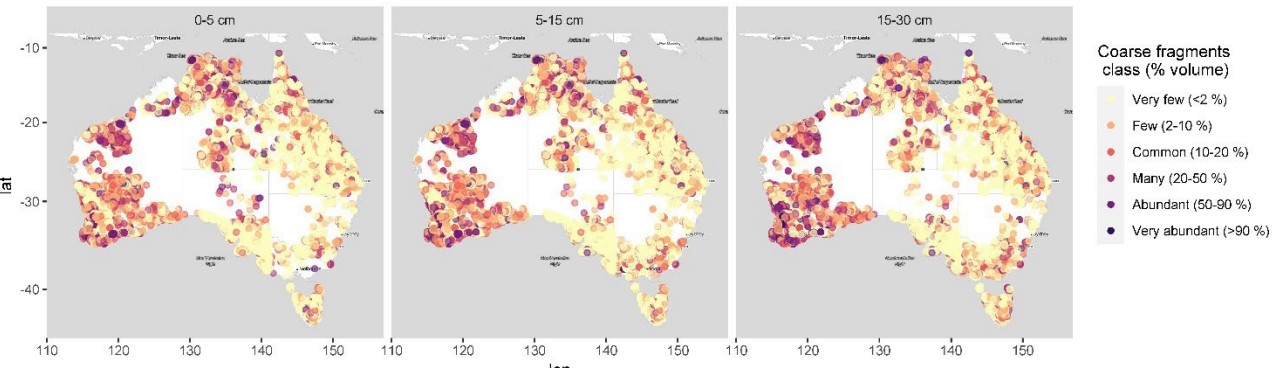

**Figure 3:Calibration data on coarse fragments (% volume) classes for the 0-5, 5-15, 15-30 cm depth intervals.**

### 2.7 Environmental covariates

We selected 61 spatially-exhaustive covariates (90 m grid cell size) for Australia made available by TERN, describing soil-forming factors and soil properties (Table 2), to calibrate *scorpan* models (Mcbratney et al., 2003). A scorpan model establishes

quantitative relationships between soil properties or classes and other soil properties (*s*), climate (*c*), organisms (*o*), relief (*r*), parent material (*p*), age (*a*), and spatial position (*n*), i.e., *soil = f (s,c,o,r,p,a,n)* (McBratney et al., 2003). We used the soil properties of clay and sand content for each GSM layer due to the relevant role of soil texture in SOC stabilization. We selected 15 climatic covariates since climate is a relevant driver of SOC storage from the global to subregional scale (Wiesmeier et al., 2019), influencing both the SOC decomposition and the C input into the soil. The organisms factor was represented by 22

covariates related to net primary productivity (e.g., Normalized Difference Vegetation Index (NDVI), enhanced vegetation index (EVI)) and vegetation type which influence the C allocation patterns, quantity and pathways of C input (e.g.,





aboveground vs. belowground). Long-term average NDVI calculated with Landsat 5 surface reflectance data was processed with Google Earth Engine. Eight relief covariates were surrogates for elevation, water flow, erosion processes, and sediment transport and accumulation, derived from the 3-second Shuttle Radar Topographic Mission (SRTM) (Farr et al., 2007). Gamma

radiometrics, total magnetic intensity and weathering intensity index (Wilford, 2012a) were used as proxies for parent material and age. Gamma radiometrics inform on regolith's mineralogy and surface geochemistry. The concentrations and ratios of three radioelements (potassium (K), thorium (Th), uranium (U)) are indicators of the lithology and degree of weathering (Wilford and Minty, 2007). The weathering intensity index map (Wilford, 2012) was generated from field-based data on the degree of bedrock weathering and multivariate analysis using gamma radiometrics and relief covariates as predictors.

The covariates were void-filled by a combination of plane fitting and inverse distance weighting. The plane fitting method was used to compute the average value of the neighbouring pixels, and otherwise an inverse distance weighting algorithm with default parameters was applied when the error in the plane fitting was large. This is the default implementation from ArcGIS software 10.5.

[Table 2 is at the end of the manuscript]

## 2.8 Modelling SOC fractions, SOC vulnerability and coarse fragments

Quantile regression forests (Meinshausen, 2006) is a generalization of the popular machine-learning algorithm random forests (Breiman, 2001). Random forests is based on an ensemble of regression trees. Each decision tree is fitted on a bootstrap sample of the original data. Further randomness is incorporated in each individual regression tree by selecting a subset of variables in

each node for which the split is made. Whereas random forests report the mean from the observations allocated in each final node, quantile regression forests keeps all values (Meinshausen, 2006), thus estimates of conditional quantiles can be made (Meinshausen, 2006). In DSM, quantile regression forests was applied previously by Vaysse and Lagacherie (2017).

We fitted quantile regression forest models for ilr1, ilr2, and Vp by GSM depth interval with the following settings: ntree = 500 (number of trees), default values of nodesize = 5 (minimum number of observations in terminal nodes) and mtry = 7

(number of predictor variables subset as candidates to make the split at each node). The models were calibrated with the *ranger* package (Wright and Ziegler, 2017) for the R environment. We fitted probability random forest models for coarse fragments (Malley et al., 2012), setting nodesize = 10. In probability random forest models, each tree predicts the class probability for each sample, and these are averaged for the forest probability estimate. The models were evaluated with 10-fold cross-validation. Variable importance was calculated with permutation (Breiman, 2001) on models fitted with 5000 trees and all

observations. After the regression trees are constructed, the values of a variable of interest are randomly permuted and the error for the out-of-bag data is estimated. The variable importance is given by the percent increase in error compared with the out-of-bag predictions leaving all variables intact.

We checked the spatial structure of the regression model residuals with cross- and auto- variograms. The spatial cross-correlation was somewhat important at a short range (approximately 20 km) for the 15-30 cm depth interval, but with a high



nugget-to-sill ratio in the fitted variogram models. For the 0-15 cm and 5-15 cm depth, only the residuals of ilr2 had some spatial structure. Overall, we considered the spatial structure of the residuals negligible for the effects of mapping at the continental scale, and hence modelled the spatial distribution of SOC fractions with quantile regression models only.

For mapping, the expected mean values of the SOC fractions, quantile regression forest models for ilr1 and ilr2 were fitted with all available observations, predicted at 90 m resolution and back-transformed into MAOC, POC and PyOC. Similarly, the mean and prediction interval for Vp were predicted with a quantile regression forest model fitted with all observations, setting the 95th and 5th percentiles as prediction interval limits at a 90% confidence level. The prediction intervals for the SOC fractions were calculated from the full conditional probability distributions of ilr1 and ilr2 inferred from the quantile regression forest models. In the model cross-validation, for each observation, a set of 500 values was generated with simple random sampling with replacement from both ilr1 and ilr2 models. Each of the 500 pairs was back-transformed into proportions of MAOC, POC and PyOC (% of SOC). The lower and upper prediction interval limits were calculated as the 5th and 95th percentiles from the empirical distribution of the 500 values of the SOC fractions. For mapping, we used 100 simulations instead of 500 to reduce the computational time. We also mapped the probability of each $CF_{vol}$ class at 90 m resolution.

**2.9 Validation statistics**

Model performance was assessed with a random 10-fold cross-validation for the ilr-variables, back-transformed SOC fractions predictions and Vp. The validation statistics included the root mean square error (RMSE), the bias or mean error (ME), the coefficient of determination ($R^2$), and Lin's concordance correlation coefficient ($\rho_c$) (Lin, 1989). For variable $z$ at a location $s_i$, the validation statistics are calculated as:

$$RMSE = \sqrt{\frac{1}{n}\sum_{i=1}^{n}\left(z(s_i) - \hat{z}(s_i)\right)^2} \tag{5}$$

$$ME = \frac{1}{n}\sum_{i=1}^{n} z(s_i) - \hat{z}(s_i) \tag{6}$$

$$R^2 = 1 - \frac{\sum_{i=1}^{n}\left(z(s_i) - \hat{z}(s_i)\right)^2}{\sum_{i=1}^{n}(z(s_i) - \bar{z})^2} \tag{7}$$

$$\rho_c = \frac{2\rho\sigma_{\hat{z}}\sigma_z}{\sigma_{\hat{z}}^2 + \sigma_z^2 + \left(\bar{\hat{z}} - \bar{z}\right)^2} \tag{8}$$

where $z(s_i)$ and $\hat{z}(s_i)$ are observed and predicted values of z at the location $s_i$ ($i = 1, \dots, n$), $\bar{z}$ and $\bar{\hat{z}}$ are the means of the observed and predicted values, respectively, $\sigma_z^2$ and $\sigma_{\hat{z}}^2$ are their respective variances, and $\rho$ is the correlation between predicted and observed values. The concordance evaluates both the accuracy and the precision of the prediction, it can range between −1 and 1, and a value closer to 1 indicates a better agreement between predictions and observations.

We assessed the validity of the uncertainty estimates with the prediction interval coverage probability (PICP) (Shrestha and Solomatine, 2006). The PICP is calculated as:

$$PICP = \frac{count(LPL_i < z(s_i) < UPL_i)}{n} \times 100 , \tag{9}$$





where $n$ is the number of observations, and the numerator the counts that an observation $z(s_i)$ fits within its prediction interval

limits. If the prediction uncertainty was correctly estimated, for example a 90% confidence level should have a PICP value

close to 90% (approximately 90% of the observed values fall within the 90% prediction interval).

The $CF_{vol}$ models were validated with 10-fold cross-validation, assigning the class with the highest probability to each

observation prediction. Using the predicted and observed class values, we computed an error matrix. The error matrix is a two-

way contingency table composed of the observed and predicted class, for all points within the validation dataset. From the

error matrix, we calculated kappa indices: overall accuracy, user's accuracy, producer's accuracy.

The overall accuracy is the fraction of locations correctly classified. It is calculated as:

$$p = \sum_{i=1}^{U} N_{UU} / N \,, \tag{10}$$

where $U$ is the total number of classes. The user's accuracy represents the fraction of the class u that is correctly classified (i.e.

mapped class $u$ in the validation dataset is also observed as class $u$):

$$p_u = \frac{N_{uu}}{N_{u+}} \tag{11}$$

Producer's accuracy is similar to the user's accuracy but is calculated on the columns marginal or the error matrix. It is the

fraction of observations $u$ for which the prediction is also class $u$. It is obtained as follow:

$$r_u = N_{uu}/N_{+u} \tag{12}$$

For more information on kappa statistics for evaluating map accuracy we refer to Congalton (1991) and Brus et al. (2011).

**2.10 Mapping SOC fraction stocks**

The expected SOC fraction density for each GSM depth interval $i$ was calculated with the following equation, using the map

of TOC concentration from v1.2.SLGA (Wadoux et al., 2022), bulk density for the whole soil (Viscarra Rossel et al., 2015),

and volumetric coarse fragments:

$$SOC_{fraction\,i\,density} \left(Mg\,C\,ha^{-1}\,cm^{-1}\right) =$$

$$SOC_{fraction\,i} \left(\frac{mg\,C_{SOC\,fraction\,i}}{mg\,C}\right) x\,TOC \left(\frac{mg\,C}{g\,soil<2\,mm}\right) x\,Bulk\,density_{whole} \left(\frac{g\,soil}{cm^3}\right) x\,(1 - Coarse\,Fragments_{vol}\,(\%/$$

$$100)) \left(\frac{soil<2\,mm}{g\,soil}\right) x\,correction\,units \left(\frac{10^8\,cm^2}{ha} \frac{Mg}{10^9\,mg}\right) \tag{13}$$

$$Coarse\,Fragments_{vol} = \sum_{u=1}^{6} CF\,probability_u \times CF\,mid_u \tag{14}$$

where $CF\,probability_u$ is the predicted probability and $CF\,mid_u$ the midpoint of the $CF_{vol}$ class $u$. The SOC stock density

may be overestimated, especially in soils with high content of rock fragments, due to using the bulk density of whole soil and

not of fine soil (Poeplau et al., 2017). We then calculated the SOC fraction stocks for the 0-30 cm depth interval using the

median predictions of soil thickness (Malone and Searle, 2020) as a constraint in areas with shallow soils (< 30 cm). We

explored differences among SOC fraction stocks (Mg C ha$^{-1}$) by biome and land use (natural or agriculture) by taking a regular

sample of 500,000 pixels across Australia.



The uncertainty of the SOC fraction density for each GSM depth interval was estimated with simulations. We generated a
sample of 500 values from the conditional probability distributions of ilr1 and ilr2 and back-transformed those into proportions
of MAOC, POC and PyOC (% of SOC). Similarly, we generated a sample of 500 TOC values with the quantile regression
model. To account for the uncertainty of the coarse fragment volume, we generated a sample of 500 values where: 1) the
number of samples within each $CF_{Vol}$ class was proportional to the predicted class probability, and 2) assuming a continuous
uniform distribution within each class. The sample for bulk density was generated assuming a normal distribution, where the
standard deviation was calculated from the prediction interval limits, sd =UPL – LPL / 2 x z, with z-score = 1.64 for a 90%
prediction interval. We calculated the SOC fraction density for each simulation and calculated the mean and upper and lower
prediction interval limits with the 95th and 5th percentiles.

We performed a variance-based, global sensitivity analysis on every pixel to identify which variables account most for the
uncertainty of SOC fractions' density and how they vary spatially. We calculated the first-order and total effects Sobol's
sensitivity indices (Saltelli et al., 2008). The first-order sensitivity index $S_i$ represents the main effect contribution of each
input factor $X_i$ (SOC fraction %, TOC, $CF_{vol}$, and bulk density) on the variance of the output of model Y (SOC fraction density).
It is calculated as the quotient between the variance of the conditional expectation of the output with respect to an input factor,
and the the unconditional variance of the output (Saltelli et al., 2008). $S_i$ ranges between 0 and 1, and a high value indicates
that factor $X_i$ is an important contributor to the variance of the output (if we knew the true value of $X_i$ we would significantly
reduce the variance of Y). The total effect index $S_{Ti}$ includes nonadditive features of the model, by accounting for the first-
order effect of $X_i$ and its interactions with other factors on the variance of Y. When $S_{Ti}$ is close to 0, the factor is non-influential
on the model output variance. A comprehensive explanation of the variance-based methods for global sensitivity analysis can
be found in Saltelli et al. (2008). We calculated the indices with the method by Martinez (2011), with the function
sobolmartinez of the sensitivity package in R (Iooss et al., 2021).

## 385   3 Results

### 3.1 Variation of the proportions of SOC fractions with depth and biome

The boxplots of the SOC fraction calibration data showed that most of the TOC was found in the MAOC fraction with a mean
(±standard deviation) of 58.8% ± 17.5%, whereas 28.2% ± 17.5% of TOC was PyOC and 13.1% ± 11.1% was POC. The
allocation of TOC into the MAOC fractions increased with depth (Figure 4), with a mean (±standard deviation) of 56% ± 17%,
59% ± 17% and 62% ± 18% at 0-5 cm, 5-15 cm and 15-30 cm respectively. Conversely, the proportion of SOC stored as POC
decreased with depth, from 15% ± 11% and 13% ± 11% in the top 0-5 cm and 5-15 cm to 11% ± 11% at 15-30 cm.  The
percentage of SOC in the PyOC fraction remained relatively constant with depth, with values around 29% ± 17% (Figure 4).
Hence, in average, carbon vulnerability decreased with depth, with Vp = 1.4 ± 98.6 at 0-5 cm, Vp = 0.5 ± 29.7 at 5-15 cm and
Vp = 0.2 ± 3.4 at 15-30 cm.




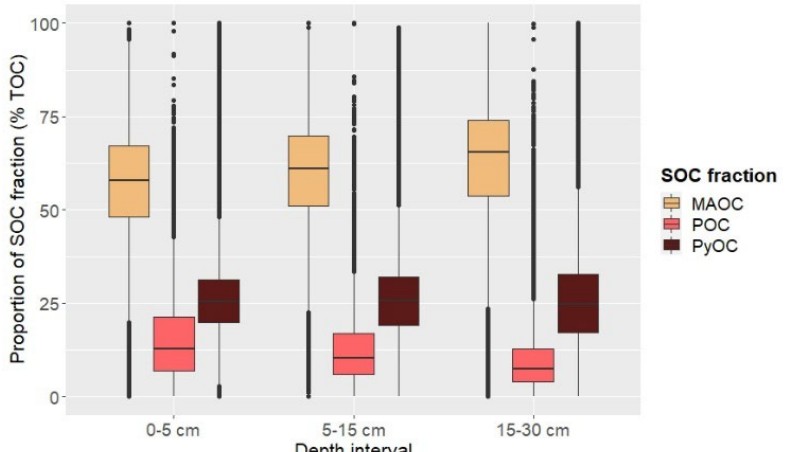

**Figure 4: Distribution of total organic carbon (TOC) among three soil organic carbon (SOC) fractions in the calibration dataset. Mineral-associated SOC (MAOC), particulate organic carbon (POC), and pyrogenic organic carbon (PyOC).**

The distribution of TOC among fractions showed similar patterns across biomes (MAOC >> PyOC > POC), except for montane grasslands and shrublands where 67% of TOC was stored as PyOC while only 27% was stored as MAOC (Table 3 and Figure S2). However, there were only 160 observations located in this biome. The proportion of TOC as POC was greater in Mediterranean forests and scrub, temperate forests, and tropical and sub-tropical forests. In Mediterranean systems, the proportion of MAOC was around 10% less than across the other biomes, (excluding montane grasslands), whereas the smallest

proportion of PyOC was found in tropical and subtropical forests. Hence, the Vp was highest in the Mediterranean biome followed by temperate forests, and tropical and subtropical forests, decreasing with depth across all biomes (Table 3).

**Table 3: Summary statistics, mean (±standard deviation) of SOC fractions and SOC vulnerability (Vp) by biome. N: number of observations. Mineral-associated SOC (MAOC), particulate organic carbon (POC), and pyrogenic organic carbon (PyOC).**

| Biome | Depth | N | MAOC (% SOC) | POC (% SOC) | PyOC (% SOC) | Vp |
|---|---|---|---|---|---|---|
| Montane Grasslands and Shrublands | 0-5 cm | 160 | 26.0 ± 9.4 | 7.1 ± 6.7 | 66.9 ± 14.3 | 0.08 ± 0.1 |
| | 5-15 cm | 160 | 26.7 ± 10.3 | 6.1 ± 5.5 | 67.2 ± 14.8 | 0.07 ± 0.1 |
| | 15-30 cm | 152 | 27.6 ± 13.0 | 4.8 ± 4.1 | 67.7 ± 16.3 | 0.05 ± 0.1 |
| Temperate Broadleaf & Mixed Forests | 0-5 cm | 2865 | 54.4 ± 10.2 | 20.6 ± 8.3 | 25.0 ± 8.1 | 2.49 ± 118.3 |
| | 5-15 cm | 2871 | 58.4 ± 10.3 | 15.6 ± 6.8 | 26.0 ± 8.5 | 0.64 ± 23.8 |
| | 15-30 cm | 2722 | 64.2 ± 11.1 | 10.3 ± 6.7 | 25.5 ± 9.8 | 0.26 ± 7.2 |
| | 0-5 cm | 750 | 57.6 ± 18.0 | 11.5 ± 7.1 | 31.0 ± 20.5 | 0.14 ± 0.1 |





| | | | | | | |
|---|---|---|---|---|---|---|
| Temperate Grasslands, Savannas & Shrublands | 5-15 cm | 762 | 60.5 ± 19.1 | 9.2 ± 6.0 | 30.5 ± 20.5 | 0.11 ± 0.1 |
| | 15-30 cm | 703 | 61.8 ± 20.1 | 7.6 ± 6.8 | 30.7 ± 21.3 | 0.09 ± 0.1 |
| Mediterranean Forests, Woodlands & Scrub | 0-5 cm | 4671 | 49.2 ± 17.0 | 20.3 ± 13.8 | 30.6 ± 20.3 | 2.46 ± 146.3 |
| | 5-15 cm | 4695 | 51.7 ± 18.3 | 17.8 ± 13.7 | 30.5 ± 20.0 | 0.97 ± 48.5 |
| | 15-30 cm | 4229 | 54.8 ± 21.0 | 15.7 ± 15.3 | 29.5 ± 20.7 | 0.28 ± 1.42 |
| Deserts & Xeric Shrublands | 0-5 cm | 2427 | 65.3 ± 16.2 | 8.3 ± 6.7 | 26.4 ± 17.6 | 0.10 ± 0.1 |
| | 5-15 cm | 2421 | 66.1 ± 16.5 | 7.5 ± 6.6 | 26.5 ± 17.7 | 0.09 ± 0.1 |
| | 15-30 cm | 2187 | 67.0 ± 17.0 | 6.7 ± 7.3 | 26.3 ± 17.7 | 0.08 ± 0.2 |
| Tropical & Subtropical Grasslands, Savannas & Shrublands | 0-5 cm | 3283 | 61.5 ± 13.9 | 10.0 ± 7.1 | 28.5 ± 14.8 | 0.12 ± 0.1 |
| | 5-15 cm | 3236 | 63.8 ± 14.5 | 8.6 ± 6.2 | 27.6 ± 15.3 | 0.10 ± 0.1 |
| | 15-30 cm | 2851 | 67.3 ± 15.0 | 7.4 ± 6.3 | 25.3 ± 15.4 | 0.09 ± 0.1 |
| Tropical & Subtropical Moist Broadleaf Forests | 0-5 cm | 242 | 61.9 ± 18.7 | 18.4 ± 11.4 | 19.7 ± 20.1 | 0.27 ± 0.4 |
| | 5-15 cm | 243 | 65.1 ± 18.7 | 16.8 ± 10.4 | 18.2 ± 19.8 | 0.24 ± 0.4 |
| | 15-30 cm | 234 | 70.1 ± 18.3 | 14.6 ± 9.6 | 15.2 ± 18.4 | 0.20 ± 0.4 |


## 3.2 Cross-validation statistics for SOC fractions

The cross-validation statistics indicated that both ilr-fractions models had similar RMSE and bias, although the $R^2$ and $\rho_c$ were better for ilr2 than ilr1. Model performance indices decreased with depth but were overall good, with $R^2 \geq 0.54$ for ilr1 and $R^2 \geq 0.68$ for ilr2. Similarly, the concordance coefficients indicated good agreement between predictions and observations, with
$\rho_c \geq 0.70$ for ilr1 and $\rho_c \geq 0.81$ for ilr2. The uncertainty was somewhat underestimated (Table 4) but close to a PICP of 90%. The cross-validation of the back-transformed SOC fractions indicated that the concordance coefficient and $R^2$ followed the trend PyOC > MAOC > POC (Table 4). Contrarily, the RMSE was greatest for MAOC with values between 8.4 % and 10.4 %, and were somewhat smaller for PyOC and POC, between 8 % and 6 %. SOC fractions predictions had a small bias, with an average underprediction of MAOC and overprediction of POC and PyOC. The accuracy plots (Supplementary material
Figure S3) indicate that the uncertainty was overestimated for MAOC and PyOC, and underestimated for POC, for all probability intervals, although the uncertainty was adequately estimated for the 90% prediction interval (Table 4). Cross-validation statistics for SOC vulnerability were worse than for the SOC fractions, with $R^2$ ranging between 0.39 to 0.56 and concordance coefficient between 0.58 and 0.72, which is possibly due to some extreme values in the calibration dataset. A PICP around 86% indicates that the uncertainty was somewhat underestimated.






**Table 4: Cross-validation statistics for ilr1, ilr2, mineral-associated SOC (MAOC), particulate organic carbon (POC), pyrogenic organic carbon (PyOC) and SOC vulnerability (Vp) by depth interval. Mean error (ME), root mean squared error (RMSE), coefficient of determination (R$^2$), Lin's concordance correlation coefficient ($\rho_c$) and prediction interval coverage probability (PICP).**

| Ilr-transformed SOC fractions | Depth | ME | RMSE | R$^2$ | $\rho_c$ | PICP (%) |
|---|---|---|---|---|---|---|
| | 0-5 cm | -0.01 | 0.45 | 0.60 | 0.75 | 85.4 |
| ilr1 | 5-15 cm | -0.01 | 0.42 | 0.63 | 0.77 | 86.1 |
| | 15-30 cm | -0.01 | 0.56 | 0.54 | 0.70 | 85.8 |
| | 0-5 cm | 0.01 | 0.46 | 0.73 | 0.84 | 86.3 |
| ilr2 | 5-15 cm | 0.01 | 0.47 | 0.74 | 0.85 | 86.5 |
| | 15-30 cm | 0.00 | 0.58 | 0.68 | 0.81 | 86.4 |
| | 0-5 cm | 0 | 0.21 | 0.39 | 0.58 | 86.1 |
| Vp | 5-15 cm | 0 | 0.17 | 0.56 | 0.72 | 86.3 |
| | 15-30 cm | 0 | 0.21 | 0.56 | 0.72 | 86.3 |
| Soil property | Depth | ME (%) | RMSE (%) | R$^2$ | $\rho_c$ | PICP (%) |
| | 0-5 cm | 1.18 | 8.42 | 0.74 | 0.85 | 95.1 |
| MAOC | 5-15 cm | 1.27 | 8.84 | 0.73 | 0.85 | 94.9 |
| | 15-30 cm | 2.00 | 10.35 | 0.68 | 0.82 | 94.9 |
| | 0-5 cm | -0.88 | 6.58 | 0.67 | 0.80 | 84.7 |
| POC | 5-15 cm | -0.83 | 6.20 | 0.65 | 0.79 | 85.5 |
| | 15-30 cm | -1.24 | 7.06 | 0.59 | 0.74 | 85.2 |
| | 0-5 cm | -0.31 | 7.72 | 0.80 | 0.89 | 93.7 |
| PyOC | 5-15 cm | -0.46 | 7.92 | 0.79 | 0.89 | 93.7 |
| | 15-30 cm | -0.77 | 8.78 | 0.76 | 0.87 | 94.0 |


### 3.3 Cross-validation statistics for coarse fragment classes

The overall accuracy of predicting coarse fragments classes was 67% for the 0-5 cm, 66% for the 5-15 cm, and 63% at the 15-30 cm depth interval. The Kappa statistics were 0.39, 0.38, and 0.37, respectively, which indicate some agreement. The producer's accuracy was around 90 % for the 'very few' class across the three depths, but the omission error was significant

for the remaining classes, especially for 'common' and 'very abundant' with values around 15 %. The user's accuracy was smaller than 50% for the classes 'few', 'common', and 'many' coarse fragments but improved somewhat for 'very abundant', 'abundant', and 'few' (Table 5). The confusion matrices are provided in the Supplementary material (Table S1-S3).





**Table 5: Cross-validation statistics for classification of coarse fragments (% volume).**

|  | Depth | Very few (< 2 %) | Few (2-10 %) | Common (10-20 %) | Many (20-50 %) | Abundant (50-90 %) | Very abundant (> 90%) |
|---|---|---|---|---|---|---|---|
| Producer's accuracy (%) | 0-5 | 93 | 39 | 17 | 35 | 22 | 15 |
|  | 5-15 | 92 | 40 | 16 | 34 | 21 | 12 |
|  | 15-30 | 92 | 38 | 13 | 31 | 39 | 8 |
| User's accuracy (%) | 0-5 | 76 | 47 | 42 | 44 | 58 | 56 |
|  | 5-15 | 74 | 47 | 43 | 43 | 53 | 63 |
|  | 15-30 | 71 | 46 | 42 | 43 | 53 | 52 |


### 3.4 Variable importance for predicting proportions of SOC fractions and coarse fragments

The environmental covariates important for both ilr1 and ilr2 models were parent material and climate, whereas relief and organisms were secondary factors (Figure 5). The rank of the 10 most important variables indicates that parent material, soil properties and relief were more relevant for ilr1, especially gravity, gamma radiometrics K, clay and elevation (DEM), whereas

for ilr2 the importance of gamma radiometrics variables and gravity was more patent for the depth intervals 5-15 cm and 15-30 cm. On the contrary, climate variables were more critical for ilr2 models, especially precipitation seasonality (PTS1 and PTS2), potential evaporation (EPA and EPX), and annual temperature (TNM and TXM), while annual temperature range (TRA) was more important for the ilr1 model. Amongst the relief covariates, DEM, topographic wetness index (TWI) and roughness were the most relevant variables for both ilr1and ilr2. Non-photosynthetic vegetation and mean EVI were the only

covariates representing the soil-forming factor organisms with some importance in the models. The most significant variables predicting the distribution of coarse fragments were gamma radiometrics and other proxies of parent material (Th, U, total dose, and ratio Th/K), followed by some covariates of climate (EPX, TRA), and relief (roughness, slope, DEM) (Figure 6).





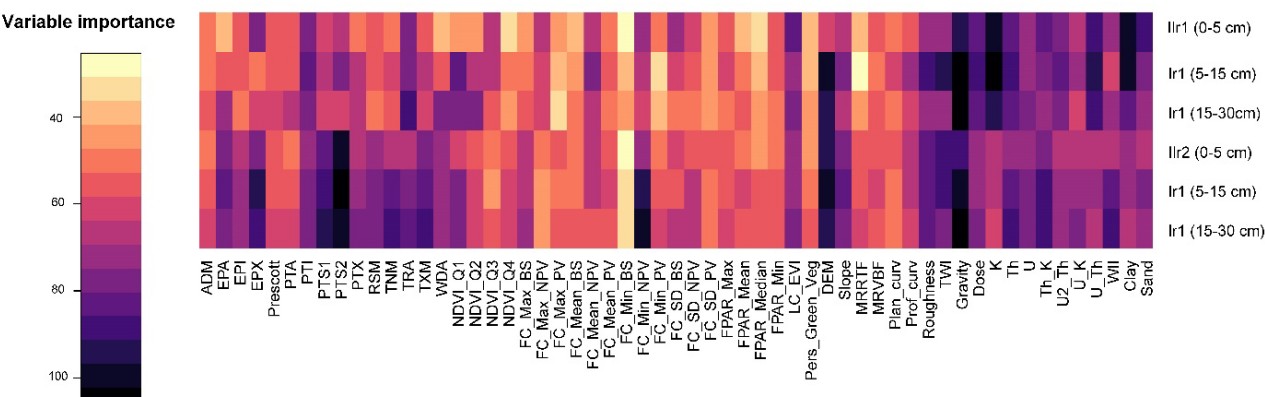

**Figure 5: Variable importance of random forest models for predicting ilr-transformed SOC fractions. Variable importance was calculated from permutation as per Breiman (2001).**

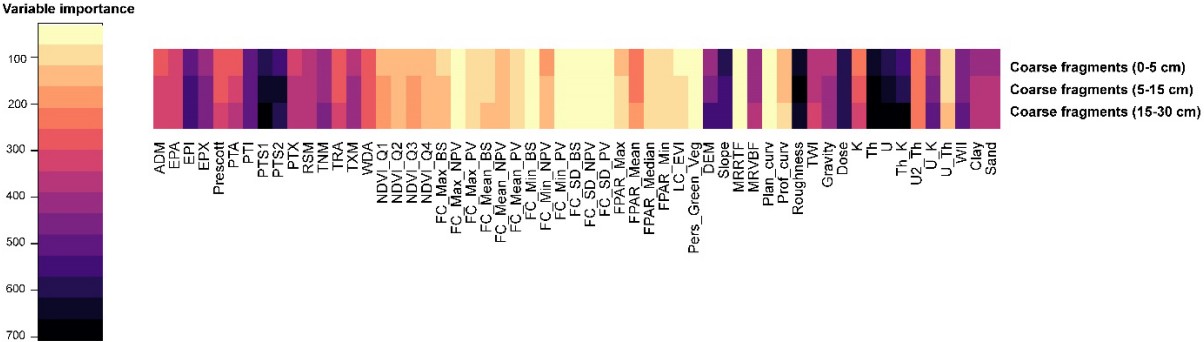

**Figure 6: Variable importance of random forest probability models for coarse fragments classes. Variable importance was estimated with Gini index impurity (Sandri and Zuccolotto, 2008).**

### 3.5 Maps of SOC fraction proportions, SOC vulnerability and coarse fragments

The spatial predictions of SOC fractions allocated most of SOC to the MAOC fraction across most of Australia (Figure 7.a and Figures S4-S6). The POC proportion was greater in the Mediterranean and temperate areas along the coast and north Queensland. There were three main regions with a high proportion of PyOC: in the north-west around Prince Regent River and Dampier Peninsula, in the west south of the Gascoyne River, and in some parts of the south-centre, e.g., east of Lake Frome. These predictions are likely driven by some calibration data with high PyOC values at these locations (Figure 2). The prediction intervals for the SOC fractions were wide (Figures S4-S6), although based on the accuracy plots (Figure S3) was estimated relatively well at the 90% confidence level. Therefore, SOC vulnerability was higher in areas with Mediterranean and temperate climate and in the centre-east (Figure 7.b). The latter is likely an artifact since that area is largely occupied by

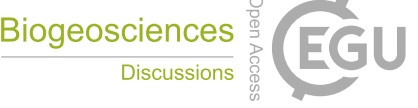

salt lakes (Kati-Thanda /Lake Eyre) and the Strzelecki desert. The coarse fragment class with highest probability across Australia was "very few" (< 2 %), and the estimated volume of coarse fragments was higher in the west and north-west Australia as well as along the Great Divide (Figures S7-S9).

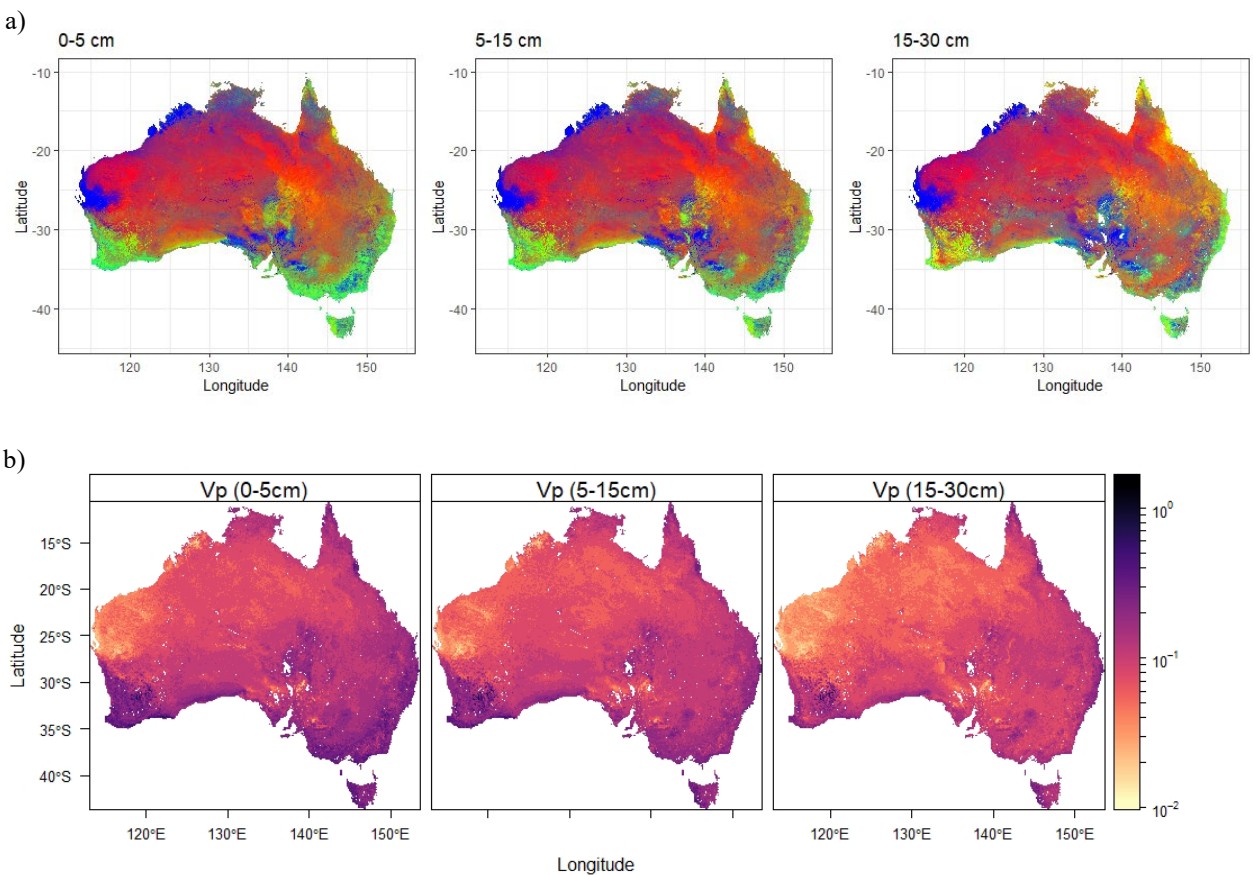

**Figure 7:a) Composite of the contribution of the three SOC fractions to SOC for the depth intervals 0-5, 5-15, 15-30 cm. The colours indicate the dominant fractions with MAOC in red, POC in green and PyOC in blue. b) SOC vulnerability for the three depth intervals. SOC vulnerability is in the log10 scale for better differentiation.**

### 3.6 SOC fraction density and sensitivity analysis

The spatial patterns of SOC fractions density are mainly determined by the spatial gradient in TOC concentration. Thus, irrespective of the dominant fraction, SOC density follows a climatic gradient and is higher in the south-west, east, Tasmania, and some regions in north Australia (Figures S10-S12). PyOC density was predicted higher along the Snowy Mountains, as the proportion of PyOC in montane grasslands and shrublands in the calibration dataset was around 67% (Table 3) and the TOC concentration was high (Figure S10). POC density was higher in southwestern Australia, east of Tasmania and in the





southeast and east. MAOC density showed similar patterns and had higher values than POC. Ultimately, the SOC fraction
stocks in the 0-30 cm (Figure 8) were not high in the east of Tasmania because these are peat soils, and the estimated median
soil thickness was estimated for mineral soils. Similarly, the SOC stocks in the Snowy mountains were constrained by the
shallow soil thickness. The total stock of SOC fractions for Australia (0-30 cm) were 12.7 Pg MAOC, 2 Pg POC and 5.1 Pg
PyOC. While we have calculated the uncertainty of the SOC fractions density (mg C cm⁻³) by depth interval with simulation

(Figures S11-S13) we have not calculated the uncertainty of the total stocks taking into account the uncertainty of soil
thickness.

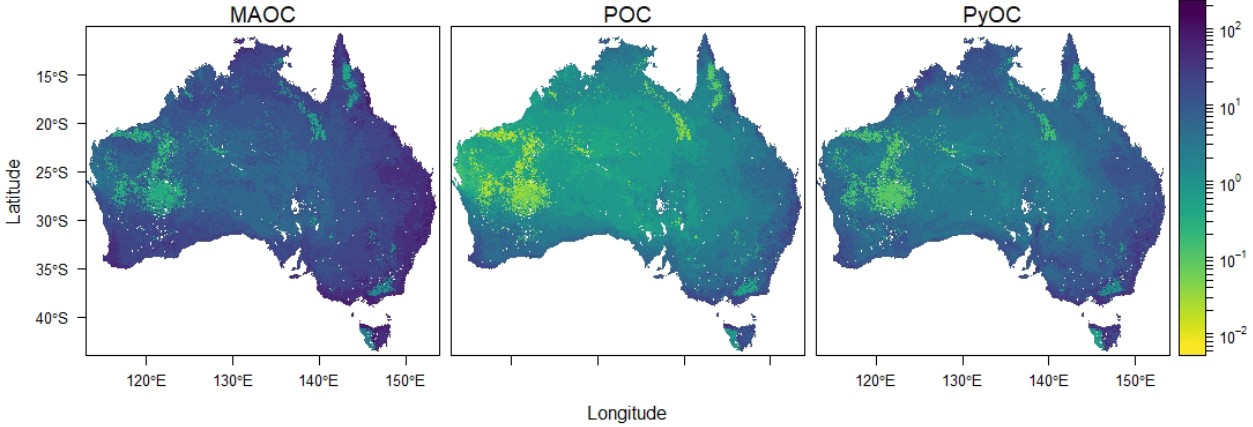

**Figure 8:SOC fraction stocks (0-30 cm) (Mg C ha⁻¹). The values are represented in the log10 scale.**

MAOC stocks were generally higher in tropical and subtropical moist broadleaf forests and temperate broadleaf and mixed
forests (Figure 9). In some biomes (temperate and tropical grasslands, savannas and shrublands, and desert and xeric
shrublands), MAOC stocks were higher in agricultural soils (which included rainfed and irrigated pastures), than in natural
areas. In the Mediterranean biome, MAOC stocks were similar in natural and agricultural regions. POC stocks followed similar
trends than MAOC, although in the Mediterranean biome, POC was greater in agricultural soils than in natural areas. Montane

grasslands and shrublands had maximum values of PyOC, although the median PyOC stocks were greater in the natural and
agricultural temperate biome, followed by tropical and subtropical forests (Figure 9).



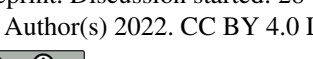

**Figure 9: SOC fraction stock (Mg C ha⁻¹) by biome and land use. Mineral-associated SOC (MAOC), particulate organic carbon (POC), pyrogenic organic carbon (PyOC).**


The two variables with most influence on the uncertainty of SOC fraction density were TOC concentration and SOC fraction % (Figures S14-S19). The total Sobol indices varied spatially depending on the SOC fraction, with TOC being the most relevant in interior areas with smaller SOC stocks, and both variables having similar influence towards the coast in the southern half of the continent. Total Sobol indices show that bulk density was not very influential on the uncertainty of the SOC fraction



density. This may be related to the relatively small prediction interval of the bulk density maps, calculated with bootstrapping
       (Viscarra Rossel et al., 2015). Coarse fragments also had a small influence as indicated by the small Sobol indices across most
       of Australia except in zones where a higher volume of coarse fragments had higher probability, e.g., western and northern
       Australia (Figures S7-S9).

## 4 Discussion

### 4.1 Differences in SOC allocation into fractions across biomes

The trend of lower MAOC proportion with increasing sand content was observed across all biomes (Figure 10). Sand content
is generally higher in the western half of Australia, coinciding with the Mediterranean and desert biomes, which have a mean
(± standard deviation) of 73% ± 14% and 71% ± 10%, respectively, in the calibration dataset. The smaller capacity of coarse-
textured soils to stabilize SOC through organo-mineral associations may partly cause the lower proportion of MAOC and
higher POC in Mediterranean soils (Figure 9 and Figure 10). Similarly, Doetterl et al. (2015) found that more SOC was stored
       as POC in arid environments where biochemical weathering is limited, due to a lower capacity for physico-chemical protection.
       Conversely, around 60% of MAOC can be found in Australian temperate grasslands, savannas and shrublands, 50 % MAOC
       in Mediterranean forests and woodlands, ~ 54-64% MAOC in temperate forests and 65-65 % MAOC in tropical (Table 3).
       Sokol et al. (2022) reported a high proportion of MAOC in temperate grasslands (~ 70 %) was due to higher NPP and microbial
decomposition favouring MAOC formation. They also found that savannas, temperate and tropical forests had a relatively high
       proportion of MAOC (~ 64 %), whereas shrublands had a lower proportion of MAOC (~ 42 %). Some differences are possibly
       explained by a higher percentage of SOC as PyOC in some systems (e.g., montane grasslands, tropical and subtropical
       savannas), in comparison to two fractions (MAOC and POC) (Sokol et al., 2022), and differences in the definition of biomes.

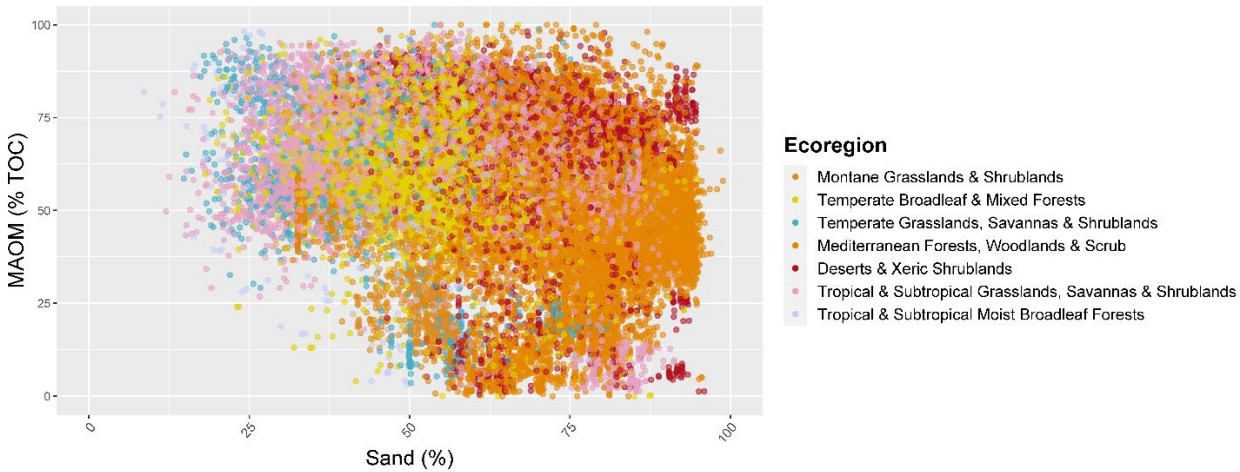




**Figure 10: Allocation of SOC into the mineral-associated organic carbon fraction (MAOC) plotted vs. sand content (%) by biome, from calibration data.**

**4.2 Abiotic and biotic predictors of SOC allocation into SOC fractions at the continental scale**

Climate and parent material were the main soil-forming factors for predicting the partition of SOC into fractions in continental
Australia. Climate is a major driver of SOC storage (Wiesmeier et al., 2019) and partition into SOC fractions at continental
scale (Bui et al., 2009). Climate influences weathering of primary minerals, the development of reactive secondary minerals
over long-term pedogenetic processes, and the chemistry of the soil solution, which in turn condition the formation of organo-
mineral associations (Kleber et al., 2015). At the same time, climate also controls SOC decomposition rates and input of
organic matter through net primary productivity. At the global scale, climate is a major driver of the abundance, persistence,
and distribution of SOC among fractions (Heckman et al., 2022), with different effects by fractions and depth. Mean annual
temperature had a strong effect on POC. In contrast, the wetness index (ratio of annual precipitation to potential
evapotranspiration) had a stronger effect on MAOC, suggesting that under wetter conditions, weathering and increasing
reactivity of minerals with depth, together with the downward transport of organic matter, enhances the formation of organo-
mineral associations (Kleber et al., 2015). Parent material alone did not have a significant effect on the partition of SOC into
MAOC at the global scale, but in interaction with the wetness index (Heckman et al., 2022).

In this study we could not verify the influence of soil geochemistry and mineralogy (metal ions, sesquioxides) on the proportion
of SOC fractions due to lack of samples with extensive laboratory analysis. Soil geochemical properties have been found to
control SOC storage at continental and regional scales (Doetterl et al., 2015b), and are involved in the stabilisation of SOC
with different mechanisms depending on the climatic context and soil pH (Lutzow et al., 2006; Rasmussen et al., 2018). In
Australia, soil physico-chemical properties, and particularly extractable iron, were the most important predictors of SOC
storage at the continental scale (Li et al., 2020). Multiple soil chemical properties can be estimated fairly well with mid-infrared
spectral models (Ng et al., 2022). Therefore, future research could expand on this study by investigating the relationships (and
their spatial patterns) between soil chemical properties (exchangeable Ca and Mg, oxalate and dithionite extractable Fe and
Al) and MAOC in the context of Australia, where the soil pH is quite acidic even under arid and semi-arid conditions.

Among vegetation variables, only EVI and the fractional cover of non-photosynthetic vegetation were important predictors for
the distribution of SOC among fractions. The fraction of non-photosynthetic vegetation may be indicative of the type of C
input into the soil (e.g., woody debris), which influences the subsequent decomposition and transformation pathways of organic
matter (Cotrufo et al., 2015) and the allocation into SOC fractions (Heckman et al., 2022).

**4.3 Vulnerability of SOC fractions to climate change**

We estimated the total stock of SOC fractions for Australia (0-30 cm) in 12.7 Pg MAOC, 2 Pg POC and 5.1 Pg PyOC, which
give a total of 19.8 Pg SOC. This value is smaller than a previous baseline of SOC stock of 25 Pg SOC for Australian topsoils,
but within its prediction interval (19-31.8 Pg C) (Viscarra Rossel et al., 2014). Differences in the total SOC stock may be partly





due to differences in TOC predictions between the previous version of SLGA v1 (Viscarra Rossel et al., 2015; Viscarra Rossel et al., 2014) and the current SLGA v1.2 (Wadoux et al., 2022). Both maps show similar patterns and range of TOC values, and hence differences in SOC fraction and total SOC stock may be mostly caused by differences in the DSM framework (e.g., calculating SOC fraction densities prior to spatialization, accounting or not with coarse fragments, etc.).

Our calculations estimated that about 64% of the total SOC is stored as mineral-associated SOC, which is consistent with other studies at global and continental scales (Heckman et al., 2022). In Australian topsoils, we estimated that only 10% of the SOC stock is stored as POC and 26% as PyOC. POC is generally more responsive than MAOC to management practices and to global change (Rocci et al., 2021). Our PyOC estimate is higher than the world average (14% of SOC (Reisser et al., 2016)) but is consistent with a study in Australia (14-33%, (Lehmann et al., 2008)). Since the 1970s, there is an upward trend in "fire weather" conditions in Australia linked to anthropogenic climate change (Harris and Lucas, 2019), which may modify the proportion and stock of PyOC.

There is great uncertainty about the effects of an increase in temperature on SOC fractions stocks and dynamics. There is large evidence supporting that the temperature sensitivity of decomposition is higher for stable SOC fractions (Conant et al., 2008; Jia et al., 2020) or SOC pools with longer mean residence time (Li et al., 2013), although other studies indicate no differences or opposite trends in sensitivity between SOC fractions (Von Lützow and Kögel-Knabner, 2009). Since most of SOC is stored as MAOC, an increase in MAOC decomposition rates with temperature (when soil moisture is not limiting) may turn some soils into a C source (Li et al., 2013). Contrarily, coarse-textured soils with a lower capacity for physico-chemical protection and a greater proportion of POC may be more vulnerable to SOC loss with an increase in temperature than fine-textured soils (Hartley et al., 2021). Heckman et al. (2022) found a decrease in SOC persistence among all SOC fractions with higher mean annual temperature at the global scale, and a decrease in SOC abundance in free POC in the surface (0-30 cm). But the increase in temperature did not affect the abundance of occluded POC and MAOC, which may be less vulnerable to warming. Similarly, a recent meta-analysis (Rocci et al., 2021) found a reduction of POC (% SOC) with warming, suggesting that soils with higher POC contribution may be more vulnerable to SOC loss. In the case of Australia, this would mean that coarse-textured soils from Mediterranean and temperate ecosystems may be more vulnerable to an increase in temperature.

Other climatic and hydrological conditions linked to climate change may also affect SOC fractions stocks in Australia. Changes in the precipitation regime (e.g., intensity and frequency of droughts, extreme precipitation events and flooding) can affect the SOC fraction stocks by either limiting or enhancing C input into the soil (effects on NPP), as well as modifying decomposition or SOC losses by increased erosion. Rocci et al. (2021) did not find clear effects on the partition of SOC among fractions with an increase in precipitation, although they found a negative tendency for POC and a positive tendency on MAOC. The effect of wind erosion on SOC loss will depend on particle size distribution and soil cover, with vulnerable soils losing 3.6 Mg C ha$^{-1}$ in south-western Australia (Harper et al., 2010). While wind erosion may deplete locally the soil of clay and silt-size particles, and light SOC fractions (light MAOC and light POC), and facilitate mineralization by disruption of aggregates, aeolian transport and deposition of may contribute to SOC enrichment in other regions (Webb et al., 2012). The influence of water erosion on SOC fractions will vary with agricultural practices, with the latter sometimes having a stronger effect on POC than





erosion depending on hillsope position (Zhao et al., 2022). SOC desestabilization and stabilization processes vary along the hillslope with changes in particle size distribution, degree of weathering, and abundance of secondary minerals (Doetterl et al., 2015a).

## 4.4 Components in the calculation of SOC fraction stocks: priorities for improving their quantification


The maps of total Sobol indices inform which variables are more influential on the uncertainty of SOC fraction densities across Australia, and can guide the priorities for mapping locally, regionally or at a continental scale the different components of SOC fraction stocks. TOC was the main variable of influence for SOC fraction density, and methods for measuring it efficiently and more economically on-farm and at the laboratory are experiencing continuous development. TOC concentration in

Australian ecosystems has been underestimated by previous SOC maps in temperate forests (Bennett et al., 2020). It is possible that in this study, the SOC fraction stocks for these ecosystems are a conservative estimate since the random forests models tend to underestimate the high values (Wadoux et al., 2022).

The percentage of TOC in the three fractions was generally the second variable of influence on the uncertainty of SOC fractions density. Still, it could be the most influential variable in areas with moderate to low SOC density. Several sources of error in

the SOC fraction predictions were not accounted for in the sensitivity analysis, like the error propagated from the different spectral models or the fact that the fractionation in the original dataset was applied to agricultural soils and some pastures but lack forest soils. There is also an under representation of some biomes and land cover types (e.g., tropical savannas) in the dataset used for fractionation. Ideally, the spectral models could be improved by increasing the representation of different natural ecosystems, which may have very different mechanisms of stabilization.

We used estimates of rock fragment volume in the calculation of SOC fraction stocks, which can overestimate the stocks when the bulk density is for the whole soil instead of that of the fine soil (< 2 mm) (Poeplau et al., 2017). The error due to combining volumetric coarse fragments and bulk density of the whole soil is not captured in the sensitivity analysis. Neglecting the content of coarse fragments can significantly overestimate the SOC fraction stocks in soils with non-negligible stoniness (> 5 %), more than doubling the actual stocks in soils with > 30% rock fragments (Poeplau et al., 2017). We anticipated that the inaccuracy

of the coarse fragments' maps and the broad range within each category would contribute significantly to the error and uncertainty in the SOC fraction stocks estimates. This was true in areas with non-negligible stoniness (> 2%), as indicated by the total Sobol indices (Figures S14-S19). However, compared with the distribution among SOC fractions and TOC concentration, coarse fragments were not the most relevant variable influencing SOC fraction density.

## Conclusions

SOC fractions are crucial as input for SOC dynamics models. These maps of MAOC, POC and PyOC can be used as input for modelling SOC stocks under different management strategies and identify areas where SOC stocks could be augmented more efficiently (i.e., areas with higher SOC sequestration potential/SOC deficit) (Meyer et al., 2017; Martin et al., 2021). Land





management practitioners could then optimize the spatial allocation of different agricultural practices while maintaining several soil functions and services, mainly food security and climate change mitigation and adaptation.

The main covariates predicting the distribution of SOC among fractions at the continental scale were identified as climate and parent material. Yet, a comprehensive and homogeneous dataset that examines the soil geochemical properties (e.g., exchangeable Ca and Mg, extractable Fe and Al, CEC ascribed to minerals, etc.) controlling SOC stabilisation processes is lacking in Australia. The diversity of climatic and pedological conditions suggests that different mechanisms will control SOC stabilization and dynamics across the continent, as observed in other regions (Rasmussen et al., 2018). The link between

mycorrhizal associations (Averill et al., 2014; Jo et al., 2019) and soil microbial community composition (e.g., $N_2$-fixing organisms), soil stoichiometry and vegetation communities (Bui and Henderson, 2013), and their effects on SOC fractions (Cotrufo et al., 2019) should be further investigated. For example, it is possible that in native ecosystems with higher soil C:N ratio and recalcitrant litter, there may be a high proportion of SOC as POC, whereas the MAOC fraction may not be C-saturated.

The uncertainty on the spatial predictions of SOC fraction stocks was driven mainly by TOC and the proportion of SOC fractions predictions, which in turn rely on spectral predictive models developed with soil samples originating mainly from agricultural soils. However, the sensitivity analysis allows targeting variables that should be prioritised at the local and regional scale to reduce the uncertainties of SOC fraction stock estimates. Future works shall include more efforts into sampling for measuring TOC, fractionation on underrepresented regions, or developing local spectral models for predicting SOC fractions.

## Acknowledgments

The authors acknowledge the Sydney Informatics Hub and the use of the University of Sydney's high performance computing cluster, Artemis, that enabled the computations for mapping at 90 m resolution for continental Australia. We also acknowledge the high-performance computing facilities from CSIRO that were used for mapping at 90 m resolution the SOC fraction densities. Some data was sourced from Terrestrial Ecosystem Research Network (TERN) infrastructure, which is enabled by

the Australian Government's National Collaborative Research Infrastructure Strategy (NCRIS). This work was partly funded by the Terrestrial Ecosystem Research Network (TERN), an Australian Government NCRIS-enabled project. MRD, AMJCW BM and AMcB acknowledge support from the Research Portfolio at the University of Sydney. AMcB acknowledges support via the Australian Research Council's Laureate Program entitled 'A calculable approach to securing Australia's soil'

## Data availability

The observed data on total organic carbon and coarse fragments are publicly available in their majority from the Soil Data Federator ([https://esoil.io/TERNLandscapes/Public/Pages/SoilDataFederator/SoilDataFederator.html](https://esoil.io/TERNLandscapes/Public/Pages/SoilDataFederator/SoilDataFederator.html)). The covariate data is available at [https://esoil.io/TERNLandscapes/Public/Pages/COGs/](https://esoil.io/TERNLandscapes/Public/Pages/COGs/). The R scripts used to develop the maps and figures for the



paper are publicly available at https://github.com/AusSoilsDSM/SLGA/tree/main/Production/DSM. The spectral models and

SOC fraction predictions are available upon reasonable request. The maps can be downloaded from

https://data.tern.org.au/landscapes/slga/NationalMaps.

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





**Table 2: List of environmental covariates with unit and associated reference when applicable. All covariates are in geographic coordinates with 3 arc second grid cell (about 90 m x 90 m) resolution with coordinate system WGS84 (EPSG:4326) and extent: 112.99958°E - 153.99958°E; 10.0004°S - 44.00042°S.**

| Factor | Covariate | Predictor variable | Unit | Reference |
|---|---|---|---|---|
| Soil | Clay | Depth-specific soil clay content (0-5 cm, 5-15 cm, 15-30 cm) | percent | Malone and Searle (2021) |
| | Sand | Depth-specific soil sand content (0-5 cm, 5-15 cm, 15-30 cm) | percent | Malone and Searle (2021) |
| Climate | ADM | Mean annual aridity index (annual precipitation/annual potential evaporation) | index | Harwood (2019) |
| | EPA | Annual potential evaporation | mm | Harwood (2019) |
| | EPI | Minimum monthly potential evaporation | mm | Harwood (2019) |
| | EPX | Maximum monthly potential evaporation | mm | Harwood (2019) |
| | Prescott | Prescott Index generated by using Prescott index = 0.445P / E0.75 | index | - |
| | PTA | Annual precipitation | mm | Harwood (2019) |
| | PTI | Minimum monthly precipitation | mm | Harwood (2019) |
| | PTS1 | Precipitation seasonality 1- solstice seasonality composite factor ratio | ratio | Harwood (2019) |
| | PTS2 | Precipitation seasonality 2- equinox seasonality composite factor ratio | ratio | Harwood (2019) |
| | PTX | Maximum monthly precipitation | mm | Harwood (2019) |
| | RSM | Short-wave solar radiation - annual mean (SRAD data) | MJ/m2/day | Harwood (2019) |
| | TNM | Minimum temperature – Annual mean | °C | Harwood (2019)) |
| | TRA | Annual temperature range | °C | Harwood (2019 |
| | TXM | Maximum temperature - Annual mean | °C | Harwood (2019) |
| | WDA | Annual atmospheric water deficit (annual precipitation – annual potential evaporation) | mm | Harwood (2019) |
| Organisms/ Vegetation | NDVI_Q1 | Landsat 5 long-term average NDVI (January-March) 1986-2011 | unitless | U.S. Geological Survey Landsat 5 Surface Reflectance Tier 1. Masek et al. (2006) |
| | NDVI_Q2 | Landsat 5 long-term average NDVI (April-June) 1986- 2011 | unitless | U.S. Geological Survey Landsat 5 Surface Reflectance Tier 1. Masek et al. (2006) |





| NDVI_Q3 | Landsat 5 long-term average NDVI (July-September) 1986-2011 | unitless | U.S. Geological Survey Landsat 5 Surface Reflectance Tier 1. Masek et al. (2006) |
|---|---|---|---|
| NDVI_Q4 | Landsat 5 long-term average NDVI (October-December) 1986 - 2011 | unitless | U.S. Geological Survey Landsat 5 Surface Reflectance Tier 1. Masek et al. (2006) |
| FC_Max_BS | Landsat Fractional cover - Bare Soil -Maximum (1987 – 2019) | percent | Joint Remote Sensing Research Program (2021) |
| FC_Max_NPV | Landsat Fractional cover - Non Photosynthetic Vegetation - Maximum (1987 – 2019) | percent | Joint Remote Sensing Research Program (2021) |
| FC_Max_PV | Landsat Fractional cover - Photosynthetic Vegetation - Maximum (1987 – 2019) | percent | Joint Remote Sensing Research Program (2021) |
| FC_Mean_BS | Landsat Fractional cover - Bare Soil - Mean (1987 – 2019) | percent | Joint Remote Sensing Research Program (2021) |
| FC_Mean_NPV | Landsat Fractional cover - Non Photosynthetic Vegetation - Mean (1987 – 2019) | percent | Joint Remote Sensing Research Program (2021) |
| FC_Mean_PV | Landsat Fractional cover - Photosynthetic Vegetation - Mean (1987 – 2019) | percent | Joint Remote Sensing Research Program (2021) |
| FC_Min_BS | Landsat Fractional cover - Bare Soil Minimum (1987 – 2019) | percent | Joint Remote Sensing Research Program (2021) |
| FC_Min_NPV | Landsat Fractional cover - Non Photosynthetic Vegetation - Minimum (1987 – 2019) | percent | Joint Remote Sensing Research Program (2021) |
| FC_Min_PV | Landsat Fractional cover - Photosynthetic Vegetation - Minimum (1987 – 2019) | percent | Joint Remote Sensing Research Program (2021) |
| FC_SD_BS | Landsat Fractional cover - Bare Soil - Standard deviation (1987 – 2019) | percent | Joint Remote Sensing Research Program (2021) |
| FC_SD_NPV | Landsat Fractional cover - Non Photosynthetic Vegetation - Standard deviation (1987 – 2019) | percent | Joint Remote Sensing Research Program (2021) |
| FC_SD_PV | Landsat Fractional cover - Bare Soil - Standard deviation (1987 – 2019) | percent | Joint Remote Sensing Research (2021) |
| FPAR_Max | Fraction of Photosynthetically Active Radiation (FPAR) - AVHRR - Maximum Value in time series | percent | Donohue et al. (2021) |
| FPAR_Mean | Fraction of Photosynthetically Active Radiation (FPAR) - AVHRR - Mean Value in time series | percent | Donohue et al. (2021) |
| FPAR_Median | Fraction of Photosynthetically Active Radiation (FPAR) - AVHRR - Median Value in time series | percent | Donohue et al. (2021) |



| | | | | |
|---|---|---|---|---|
| | FPAR_Min | Fraction of Photosynthetically Active Radiation (FPAR) - AVHRR - Minimum Value in time series | percent | Donohue et al. (2021) |
| | LC_EVI | National Dynamic Land Cover Dataset Mean of enhanced vegetation index (EVI) for the timeseries 2000 – 2008 | unitless | Lymburner et al. (2010) |
| | Pers_Green_Veg | Landsat 2000-2010 Persistent Green-Vegetation Fraction | unitless | Johansen et al. (2012) |
| Relief | DEM | Elevation 3 Second - Shuttle Radar Topography Mission | meter | Farr et al. (2007) |
| | MRRTF | Multi-resolution Ridgetop Flatness | unitless | Gallant and Austin (2015) |
| | MRVBF | Multiresolution Index of Valley Bottom Flatness | unitless | Gallant and Dowling (2003) |
| | Plan_curv | Plan curvature | unitless | Wilson and Gallant (2000) |
| | Prof_curv | Profile curvature | unitless | Wilson and Gallant (2000) |
| | Roughness | Relief roughness | unitless | Wilson and Gallant (2000) |
| | Slope | Slope | percent | Zevenbergen and Thorne (1987) |
| | TWI | Topographic wetness index | unitless | Wilson and Gallant (2000) |
| Parent material/ Age | Gravity | Gravity Anomaly Grid of Australia | unitless | Lane et al. (2020) |
| | Dose | Radiometric grid of Australia (Radmap) v4 2019 - Filtered dose | unitless | Wilford and Kroll (2020) |
| | K | Radiometric grid of Australia (Radmap) v4 2019 - Potassium | percent | Wilford and Kroll (2020) |
| | Th | Radiometric grid of Australia (Radmap) v4 2019 - Thorium | ppm | Wilford and Kroll (2020) |
| | U | Radiometric grid of Australia (Radmap) v4 2019 - Uranium | ppm | Wilford and Kroll (2020) |
| | Th_K | Radiometric grid of Australia (Radmap) v4 2019 - Thorium Potassium ratio | ratio | Wilford and Kroll (2020) |
| | U2_Th | Radiometric grid of Australia (Radmap) v4 2019 - Uranium squared to Thorium ratio | ratio | Wilford and Kroll (2020) |
| | U_K | Radiometric grid of Australia (Radmap) v4 2019 - Uranium Potassium ratio | ratio | Wilford and Kroll (2020) |
| | U_Th | Radiometric grid of Australia (Radmap) v4 2019 - Uranium Thorium ratio | ratio | Wilford and Kroll (2020) |
| | WII | Weathering intensity index | unitless | Wilford (2012b) |
