# Peer review of "Mapping soil organic carbon fractions for Australia, their stocks and uncertainty"

_Biogeosciences, 2022_

## Referee Comment (RC1)

**Reviewer 2:**
**Manuscript Number: bg-2022-207**

**General Comments**

This manuscript and the associated data provide a high value set of SOC fraction gridded products for Australia and will be an important resource for land managers and the scientific community. There are some question marks over the reliability of the data due to bias in the original calibration samples (primarily agricultural soils) and in the method used to derive the PyOC and MAOC fractions for the calibration samples. The approach used can clearly be made more robust in future by starting with a larger, more representative and more reliable set of calibration data.

**Addressing the review criteria:**

1. Does the paper address relevant scientific questions within the scope of BG?
   Yes

2. Does the paper present novel concepts, ideas, tools, or data?
   Yes. The publication and data will be of considerable use to management and science communities in Australia and globally.

3. Are substantial conclusions reached?
   Yes.

4. Are the scientific methods and assumptions valid and clearly outlined?
   This are some weaknesses in the scientific methods/sampling that are the basis of this manuscript. The authors point out issues in the sampling, where there was insufficient range of soil types/biomes used in the initial calibration. What they do not point out is that the method used to estimate the fractions is not particularly reliable.

   The SOC fractions used for calibration of the spectral methods were measured using a chemical approach (and data) developed for SCaRP a decade ago. The $^{13}$C NMR based approach to determine PyOC is not widely used for estimating concentrations/stocks as it is a semi-quantitative method (e.g it often under-detects aromatic C and spectral assignment /integration is difficult in low-SOC samples).
   The current global data set of PyOC (black carbon) has been obtained using the Benzene PolyCarboxylic Acid (BPCA) approach, this approach gives a more realistic estimate of concentrations of PyOC in soil (see Jones, A. et al. "Fires prime terrestrial organic carbon for riverine export to the global oceans," Nat. Commun. 11, 2791 (2020). https://doi.org/10.1038/s41467-020–16576-z', Dymov, A.A. et al. Comparison of the Methods for Determining Pyrogenically Modified Carbon Compounds. *Eurasian Soil Sc.* **54**, 1668–1680 (2021). https://doi.org/10.1134/S1064229321110065 )

5. Are the results sufficient to support the interpretations and conclusions?
   Yes.

6. Is the description of experiments and calculations sufficiently complete and precise to allow their reproduction by fellow scientists (traceability of results)?
   Yes.

7. Do the authors give proper credit to related work and clearly indicate their own new/original contribution?
   Yes.

8. Does the title clearly reflect the contents of the paper?
   Yes.

9. Does the abstract provide a concise and complete summary?
   Yes.

10. Is the overall presentation well structured and clear?
    Yes.

11. Is the language fluent and precise?
    Yes.

12. Are mathematical formulae, symbols, abbreviations, and units correctly defined and used?
    Yes.

13. Should any parts of the paper (text, formulae, figures, tables) be clarified, reduced, combined, or eliminated?
    No.

14. Are the number and quality of references appropriate?
    Yes.

15. Is the amount and quality of supplementary material appropriate?
    Yes.

**Specific Comments**

**2 Materials and Methods**

1. PyOC was estimated by $^{13}$C CP MAS NMR, this is excellent for identifying the types of carbon present but is generally regarded as only semi-quantitative in nature. The data for the fractions have not been cross-correlated with other approaches such as BPCA to ensure that they are in fact robust. MAOC is estimated by difference subtracting POC and PyOC from SOC, as a result there is also a question mark over the reliability of the estimates of this variable.
What is needed are measures of the error associated with PyOC and MAOC and how this transfers to errors in the subsequent estimates of these fractions using the irl1 and irl2 models.

2. The process of generating MAOC, POC and PyOC is reliant on an initial calibration and this was done on 312 samples a decade ago. A much larger number of samples were used in the spectral harmonisation and data set modelling but in the end they are reliant on this small number of samples (312) for SCaRP and an even smaller number of SCaRP samples (200) for the AusSpecMIR and AusSpecMIR2 and 309 for AusSpecIR. Again the authors need to justify how the use of this small set of samples, dominated by agricultural soil types, is able to successfully be used to provide calibration data for the much wider set of biome/soil types used in this study.

This may help to explain why the authors experienced difficulty with their preferred approach of modelling SOC fraction concentrations directly: '*The Pearson's r correlation coefficient 220 was 0.56, but the sum of SOC fractions showed some extreme values (Figure S1).*

3. The authors have used a very thorough and well thought through approach to generate a spatially and depth consistent gridded set of SOC fraction data for the continent.
Figure 2 (I think) shows where the spectral training data sets were located. It would be useful to see where the original SCaRP SOC fraction calibration data was collected, either on a map of Australia or in a table by soil/biome type, this would provide the reader with a clearer idea of the limitations due to type of calibration data used.

**3 Results**

4. The reporting of errors is an issue in the manuscript.

The authors need to do a major check over all sections of the text and tables to ensure that precision is treated correctly.  Errors should have 1 or at most 2 significant digits.
The problems start in the abstract, *59% ±17.5%, whereas 28% ± 17.5% was PyOC and 13% ± 11.1%*
in this case the errors have more decimal places than the values and there are too many significant digits for such large errors.
The estimate of stocks *12.7 Pg MAOC, 2 Pg POC* has inconsistent precision, possibly this is correct but given the other issues possibly not.
As a clearer example the authors report (L388) *13.1% ± 11.1%,* this should be 13 +/- 11 which indicates 85% error, reporting to +/- 0.1 (0.7%) clearly makes little sense.
The decimal place in the error sets the decimal place in the value, they should always agree.
Things get worse : Table 3.  *2.49 +/- 118.3     0.64 +/- 23.8*

5. It would be very useful to have stocks with errors estimated based on the data generated from the grid.  The authors estimate stocks but then they provide no errors because of an issue around soil thickness.  Without an error then these stocks are of limited use (see above).
It would be better to have some estimates of error for these stocks that sum over the best estimates for soil thickness, issues around soil bulk density estimates etc.

**4 Discussion**

6. Mention is made of a likely underestimate of SOC in forest systems and it is clear the authors were aware of the lack of calibration sampling in forest systems. *'or the fact that the fractionation in the original dataset was applied to agricultural soils and some pastures but lack forest soils.' 'The uncertainty on the spatial predictions of SOC fraction stocks was driven mainly by TOC and the proportion of SOC fractions predictions, which in turn rely on spectral predictive models developed with soil samples originating mainly from agricultural soils.'*
Australia has 134 million hectares of forest, 17% of the land surface area.
https://www.agriculture.gov.au/abares/forestsaustralia/australias-forests#forest-area
It is somewhat surprising that they did not add additional calibration samples from the forest estate into the early stages of this study. The potential magnitude of this underestimation might be mentioned.

**Technical Comments**

Line 73
In Australia, the long history of burning suggests pyrogenic organic carbon (PyOC) is an additional important component (Lehmann et al., 2008) of SOC. PyOC refers to charred residues derived from

Line 75
comprised by a continuum of organic compounds thermally altered by fire

Line 90
the contribution of SOC fractions (MAOC, POC, and PyOC) to the total SOC in the top 30 cm of the soil  and update the Soil

Line 101
mapped proportions of SOC. Finally, we calculated SOC fraction stocks for the 0-30 cm topsoil using data from SLGA maps

Line 135
The content of poly-aryl C

Line 164
resolution. Soil samples from AusSpecMIR and AusSpecMIR2 were scanned in quadruplicate.

Line 180
soil), total organic carbon (TOC) concentration (mg C g$^{-1}$ soil), and used harmonised spectra.

Line 241
The Soil Data Federator is a web API that compiles soil data from different institutions

Line 292
(Meinshausen, 2006). In Digital  Soil Mapping (DSM), quantile regression

Line 445
for ilr2 the importance of gamma radiometrics variables and gravity was more important for the

---

## Referee Comment (RC2)

**Reviewer: 1**

**Manuscript Number: bgb-2022-207**

**General Comments**

The paper presented by Dobarco and colleagues represents an ambitious undertaking to map the fractional contribution of three different organic matter fractions (MAOM, POM, and PyOM), building on previous work from Grundy et al. (2015) and Viscarra Rossel et al., (2019). Using fractional carbon data predicted via mid- and near- infrared spectroscopy and quantile regression random forest model, they produce a useful gridded dataset of MAOC, POC, and PyOC to a depth of 30 cm. Generally, I found the paper to be well-written and insightful and have only few comments which I detail below.

| Principal Criteria | Excellent (1) | Good (2) | Fair (3) | Poor (4) |
|---|---|---|---|---|
| **Scientific significance:** Does the manuscript represent a substantial contribution to scientific progress within the scope of Biogeosciences (substantial new concepts, ideas, methods, or data)? | | X | | |
| **Scientific quality:** Are the scientific approach and applied methods valid? Are the results discussed in an appropriate and balanced way (consideration of related work, including appropriate references)? | X | | | |
| **Presentation quality:** Are the scientific results and conclusions presented in a clear, concise, and well-structured way (number and quality of figures/tables, appropriate use of English language)? | | X | | |

**Review Criteria**

1. Does the paper address relevant scientific questions within the scope of BG?
   Yes.
2. Does the paper present novel concepts, ideas, tools, or data?
   Yes, especially the published gridded data.
3. Are substantial conclusions reached?
   Yes, the spatial distribution of POM, MAOM, and PyOM across Australia presents a substantial step forward towards understanding controls on SOM formation.
4. Are the scientific methods and assumptions valid and clearly outlined?
   Yes.
5. Are the results sufficient to support the interpretations and conclusions?
   Yes.
6. Is the description of experiments and calculations sufficiently complete and precise to allow their reproduction by fellow scientists (traceability of results)?
   Yes.

7. Do the authors give proper credit to related work and clearly indicate their own new/original contribution?
Yes.
8. Does the title clearly reflect the contents of the paper?
Yes.
9. Does the abstract provide a concise and complete summary?
Yes.
10. Is the overall presentation well-structured and clear?
Yes.
11. Is the language fluent and precise?
Generally, yes.
12. Are mathematical formulae, symbols, abbreviations, and units correctly defined and used?
Yes.
13. Should any parts of the paper (text, formulae, figures, tables) be clarified, reduced, combined, or eliminated?
Yes.
14. Are the number and quality of references appropriate?
Yes.
15. Is the amount and quality of supplementary material appropriate?
Yes.

**Line Comments**

Lines 41 – 43: It's not clear whether SOM or SOC is being discussed in this sentence. I think the sentiments expressed are true in both instances, but it's worth rephrasing to improve clarity.

Lines 47 – 49: Is this true for PyOC as well? What is the primary mechanism of preservation for pyrogenic organic matter if not some level of biochemical recalcitrance?

Line 74: In the MAOM fraction, what is the mechanism of preservation of PyOM in the MAOM fraction? Is it occlusion within microaggregates that are smaller than 60/53 µm? Or can PyOM form organo-mineral association? I think it this section it would be worth discussing this fraction in a little more depth.

Lines 77 – 78: This is kind of what I mean in my comment above, it seems contradictory.

Lines 88 – 89: It would be great to have a citation here for either how fractions can inform management, or how they can be incorporated into policy.

Lines 139-142: Could you add a discussion either here or later in section 2.3 related to the pre-processing of the data from the different libraries? Were the data smoothed and corrected using Savitzky-Golay or the like, and how did that differ across the different libraries? If they are raw spectra, please specify that they were received in that form.

Line 219-220: Are there data associated with the C recovery of the fractions that might explain some of the poor matching between TOC and fraction sums? Soluble and dissolved carbon can be lost throughout the fractionation process, which may bias predictions.

Line 235: Can you clarify the difference between ilr1 and ilr2 in this line? Looking through the equations and text I think I can piece it together, but it would be helpful to the reader to make it explicit.

Line 280: How much of the dataset was void-filled? Can you provide a percentage for total data interpolation across covariates?

Line 387 and on: As my fellow reviewer noted, there are inconsistencies in the reporting of significant digits and errors in the results section of the manuscript. At the risk of being redundant, I recommend the authors carefully check the figures they present for consistency and utility.

Line 498-499: Are most agricultural lands in the Mediterranean biome irrigated? It could be worth highlighting the proportion irrigated either here or in the discussion.

Line 517: Higher mean sand content or higher mean SOC concentration?

Line 522-523: I think this sentence is confusing, I recommend rewording. Currently it reads as if the authors are discussing the total proportion of MAOC across Australia.

Line 541: Please clarify the directionality of the relationship between POC and MAT.

Lines 552 - 554: I'm glad you mention this here -- I was going to recommend something along these lines as a justification for not including these co-variates.

---

## Author Comment (AC2)

Manuscript Number: bg-2022-207

Responses to Reviewer 2

This manuscript and the associated data provide a high value set of SOC fraction gridded products for Australia and will be an important resource for land managers and the scientific community. There are some question marks over the reliability of the data due to bias in the original calibration samples (primarily agricultural soils) and in the method used to derive the PyOC and MAOC fractions for the calibration samples. The approach used can clearly be made more robust in future by starting with a larger, more representative and more reliable set of calibration data.

Thank you very much for the positive reception of our manuscript and your constructive comments. We will try to address the limitations of the current study and include some recommendations for improving the digital soil maps of SOC fractions in the future.

This are some weaknesses in the scientific methods/sampling that are the basis of this manuscript. The authors point out issues in the sampling, where there was insufficient range of soil types/biomes used in the initial calibration. What they do not point out is that the method used to estimate the fractions is not particularly reliable.

The SOC fractions used for calibration of the spectral methods were measured using a chemical approach (and data) developed for SCaRP a decade ago. The $^{13}$C NMR based approach to determine PyOC is not widely used for estimating concentrations/stocks as it is a semi-quantitative method (e.g it often under-detects aromatic C and spectral assignment /integration is difficult in low-SOC samples). The current global data set of PyOC (black carbon) has been obtained using the Benzene PolyCarboxylic Acid (BPCA) approach, this approach gives a more realistic estimate of concentrations of PyOC in soil (see Jones, A. et al. "Fires prime terrestrial organic carbon for riverine export to the global oceans," Nat. Commun. 11, 2791 (2020). https://doi.org/10.1038/s41467-020–16576-z', Dymov, A.A. et al. Comparison of the Methods for Determining Pyrogenically Modified Carbon Compounds. *Eurasian Soil Sc.* **54**, 1668–1680 (2021). https://doi.org/10.1134/S1064229321110065 )

Thank you for your comment. In this study we capitalized on legacy soil datasets (SCaRP) and spectral libraries to produce a new set of SOC fraction maps with digital soil mapping methods, and as it is sometimes the case when using legacy soil datasets, the laboratory method analyses may not be the most cutting-edge but are still valid for the purposes of the study.

In the original publication by Baldock et al. (2013) the terminology referred to resistant organic carbon (ROC) instead of to pyrogenic organic carbon (PyOC). ROC has a chemical composition that is not incompatible with that of charcoal (or that is dominated in its majority by charcoal and charred plant residuals) but there is a potential presence of other poly-aryl carbon compounds that do not have a pyrogenic origin (Baldock et al., 2013). We decided to change the terminology from the publication by Baldock, which referred to particulate organic carbon (POC), humus OC (HOC) and ROC, to the terminology that is currently used by most SOC fractions' studies, although we may have incurred into some imprecisions with the terms. While POC and HOC have a clear correspondence with POC and MAOC, it is possible that ROC is not completely analogous with PyOC. We will indicate this in the revised version of the manuscript, as well as recommendations for a comparison with other methods for determination of PyOC in future studies in Australia.

Baldock, J.A., Sanderman, J., Macdonald, L.M., Puccini, A., Hawke, B., Szarvas, S., McGowan, J., 2013. Quantifying the allocation of soil organic carbon to biologically significant fractions. Soil Research 51, 561-576.

**Specific Comments**

**Materials and Methods**

1. PyOC was estimated by $^{13}$C CPMAS NMR, this is excellent for identifying the types of carbon present but is generally regarded as only semi-quantitative in nature. The data for the fractions have not been cross-correlated with other approaches such as BPCA to ensure that they are in fact robust.

Thank you for your comment. As you mentioned before, we are using the legacy soil dataset from the SCaRP programme as basis for our study, and it was out of the scope of this paper, as well as far beyond our reach to perform additional fractionation analyses on the archive samples. While there may be some shortcomings associated to the determination of PyOC (or ROC) with $^{13}$C NMR, in the comparison by Dymov et al. (2021) there seems to be a good correlation between this method and the preferred BPCA (r = 0.88, p < 0.05), so we still consider it appropriate for the determination of PyOC. We will indicate in the revised manuscript that future studies should carry a comparison between PyOC determined with BCPA and $^{13}$C NMR.

MAOC is estimated by difference subtracting POC and PyOC from SOC, as a result there is also a question mark over the reliability of the estimates of this variable.
What is needed are measures of the error associated with PyOC and MAOC and how this transfers to errors in the subsequent estimates of these fractions using the irl1 and irl2 models.
There is an important clarification to make about the SOC fractionation protocol which has led to confusion on how the three fractions are quantified, and we thank the reviewer for pointing that out. We clarify the description of the SOC fractionation scheme in the revised version of the manuscript. MAOC is not estimated as SOC – (PyOC + POC). The first step of the fractionation protocol separates SOC by size. Next, $^{13}$C NMR analysis were performed for both the fine (<50 μm) and the coarse (>50 μm) fractions to determine the proportion of poly-aryl C. The content of POC and MAOC were computed as the non-PyOC SOC present in the coarse and fine fraction respectively. We have rewritten this section as follows:

"A 10-g aliquot of air-dried soil ≤ 2 mm was dispersed with 5 g L$^{-1}$ sodium hexametaphosphate and separated into coarse (>50 μm) and fine (<50 μm) fractions with wet sieving using an automated sieve shaker system (Baldock et al., 2013c). The TOC concentrations of the coarse and fine fractions were analysed with high-temperature oxidative combustion after the removal of inorganic carbon with 5-6 % H$_2$SO$_3$ if carbonates were present (method 6B3a, Rayment and Lyons (2011)). Solid-state $^{13}$C nuclear magnetic resonance ($^{13}$C NMR) spectroscopy analyses were conducted on both the coarse (>50 μm) and fine (<50 μm) fractions. $^{13}$C NMR is a semi-quantitative method that is commonly used to measure the proportion of aromatic C compounds in soil and organic matter samples. The proportion of poly-aryl C and aryl C that could be defined as lignin was determined and used as an estimate of PyOC. We note that whereas the chemical signature of the poly-aryl C is consistent with, and likely dominated by charcoal and charred plant residues, it may also indicate the presence of compounds non-pyrogenic origin (Baldock et al., 2013c). POC and MAOC contents (mg C-fraction g$^{-1}$ soil) were calculated by subtracting the proportion of PyOC in each fraction with the following equations (Baldock et al., 2013c):

$$POC = (2000 - 50 \text{ μm OC})(1 - fPyOC_{2000})MF_{2000} \tag{1}$$

$$MAOC = (\leq 50 \text{ μm OC})(1 - fPyOC_{50})MF_{50} \tag{2}$$

$$PyOC = (2000 - 50 \text{ μm OC})(fPyOC_{2000})MF_{2000} + (\leq 50 \text{ μm OC})(fPyOC_{50})MF_{50} \tag{3}$$

where 2000-50 μm OC is the measured TOC content in the coarse fraction (mg C g$^{-1}$ 2000-50 μm fraction), fPyOC$_{2000}$ is the proportion of TOC attributed to poly-aryl C in the coarse fraction, MF$_{2000}$ is the proportion of soil mass found in the coarse fraction (g 2000-50 μm fraction /g ≤ 2mm soil), ≤ 50 μm OC is the measured TOC content in the fine fraction (mg C g$^{-1}$ ≤ 50 μm fraction, fPyOC$_{50}$ is the proportion of TOC attributed to poly-aryl C in the fine fraction, and MF$_{50}$ is the proportion of soil mass found in the fine fraction (g ≤50 μm fraction /g ≤ 2mm soil)."

The propagation of error from the laboratory fractionation, spectral predictive modelling, and digital soil mapping steps is out of the scope of this paper, but we will indicate these sources of uncertainty in the discussion.

2. The process of generating MAOC, POC and PyOC is reliant on an initial calibration and this was done on 312 samples a decade ago. A much larger number of samples were used in the spectral harmonisation and data set modelling but in the end they are reliant on this small number of samples (312) for SCaRP and an even smaller number of SCaRP samples (200) for the AusSpecMIR and AusSpecMIR2 and 309 for AusSpecIR. Again the authors need to justify how the use of this small set of samples, dominated by agricultural soil types, is able to successfully be used to provide calibration data for the much wider set of biome/soil types used in this study.

This may help to explain why the authors experienced difficulty with their preferred approach of modelling SOC fraction concentrations directly: '*The Pearson's r correlation coefficient 220 was 0.56, but the sum of SOC fractions showed some extreme values (Figure S1)*'.

Thanks for your comment. We agree with the reviewer that this is one of the limitations of the study and we indicate is as such in the revised version of the manuscript. We also justify the use of the calibration dataset because the 312 samples, despite coming mainly from agricultural soils, were representative of a range of TOC content (1.2 - 90.9 mg C / g soil) and different soil types and biomes.

3. The authors have used a very thorough and well thought through approach to generate a spatially and depth consistent gridded set of SOC fraction data for the continent.

Figure 2 (I think) shows where the spectral training data sets were located. It would be useful to see where the original SCaRP SOC fraction calibration data was collected, either on a map of Australia or in a table by soil/biome type, this would provide the reader with a clearer idea of the limitations due to type of calibration data used.

Thank you for your comment. Figure 1 indicates the location of the spectral datasets (calibration data and predicted samples), including SCaRP (in red), and in Figure 2, the predictions of the spectra models at these same locations are used as calibration data for the digital soil mapping process. We have included an additional panel in Figure 1 indicating the location of the 312 SCaRP samples that were subject to fractionation at the laboratory. In addition, we have changed the caption of Figure 2 as "Location of the spectral predictions standardized for the depths 0-5 cm, 5-15 cm, and 15-30 cm, which were used as calibration data for digital soil mapping."

**Results**

4. The reporting of errors is an issue in the manuscript.

The authors need to do a major check over all sections of the text and tables to ensure that precision is treated correctly. Errors should have 1 or at most 2 significant digits.

The problems start in the abstract, 59% ±17.5%, whereas 28% ± 17.5% was PyOC and 13% ± 11.1% in this case the errors have more decimal places than the values and there are too many significant digits for such large errors. The estimate of stocks 12.7 Pg MAOC, 2 Pg POC has inconsistent precision, possibly this is correct but given the other issues possibly not.

As a clearer example the authors report (L388) 13.1% ± 11.1%, this should be 13 +/- 11 which indicates 85% error, reporting to +/- 0.1 (0.7%) clearly makes little sense. The decimal place in the error sets the decimal place in the value, they should always agree.

Things get worse : Table 3. 2.49 +/- 118.3 0.64 +/- 23.8

We apologize for this issue and have corrected the reporting of errors with the appropriate number of decimal places through the manuscript.

5. It would be very useful to have stocks with errors estimated based on the data generated from the grid. The authors estimate stocks but then they provide no errors because of an issue around soil thickness. Without an error then these stocks are of limited use (see above). It would be better to have some estimates of error for these stocks that sum over the best estimates for soil thickness, issues around soil bulk density estimates etc.

Thank you for this suggestion. We have performed 500 simulations in a subset of pixels across Australia to incorporate the uncertainty of soil thickness, bulk density, coarse fragments, TOC concentration, and distribution of Soc among fractions in the estimates of the SOC stocks for 0-30 cm depth. It was not possible to generate maps for all Australia at 90 m resolution due to time constraints for carrying the simulations, but the results inform on the variation and uncertainty of SOC stocks spatially and across biomes. We hope this additional analysis addresses the concern of the reviewer.

**Discussion**

6. Mention is made of a likely underestimate of SOC in forest systems and it is clear the authors were aware of the lack of calibration sampling in forest systems. '*or the fact that the fractionation in the original dataset was applied to agricultural soils and some pastures but lack forest soils*.' '*The uncertainty on the spatial predictions*

*of SOC fraction stocks was driven mainly by TOC and the proportion of SOC fractions predictions, which in turn rely on spectral predictive models developed with soil samples originating mainly from agricultural soils.*'
Australia has 134 million hectares of forest, 17% of the land surface area.
https://www.agriculture.gov.au/abares/forestsaustralia/australias-forests#forest-area
It is somewhat surprising that they did not add additional calibration samples from the forest estate into the early stages of this study. The potential magnitude of this underestimation might be mentioned.
Thank you for your comment. We agree with the reviewer on the need for sampling and analysing additional samples from natural systems (forests, woodlands, shrublands, etc.) in future studies. We tried to indicate this limitation in the discussion but we may not have been explicit enough, and we will emphasize it in the revised manuscript. We wish that we had more samples from forested areas, but that was out of our reach since we used legacy soil datasets and spectral libraries. We include this as a recommendation for future studies.

**Technical Comments**

We thank the reviewer for these corrections. We have incorporated them in the revised version of the manuscript.

---

## Author Comment (AC3)

Responses to Reviewer: 1
Manuscript Number: bgb-2022-207

General Comments

The paper presented by Dobarco and colleagues represents an ambitious undertaking to map the fractional contribution of three different organic matter fractions (MAOM, POM, and PyOM), building on previous work from Grundy et al. (2015) and Viscarra Rossel et al., (2019). Using fractional carbon data predicted via mid- and near- infrared spectroscopy and quantile regression random forest model, they produce a useful gridded dataset of MAOC, POC, and PyOC to a depth of 30 cm. Generally, I found the paper to be well-written and insightful and have only few comments which I detail below.

Thank you very much for your comment. We are very glad that both reviewers considered overall the paper a good contribution.

Line Comments

Lines 41 – 43: It's not clear whether SOM or SOC is being discussed in this sentence. I think the sentiments expressed are true in both instances, but it's worth rephrasing to improve clarity.
Thanks for your comment. We agree with the reviewer that these characteristics apply to both SOC and SOM, although the variety of chemical composition and organic compounds may be more pertinent for SOM, and in these organic molecules C is just one component (although the origin of that SOC is the same of the SOM it is part of, etc.). But regarding the turnover time and temporal dynamics we are thinking more of SOC. We have rephrased as follows:

*Soil organic matter consists of a continuum of compounds with different chemical compositions, origin (aboveground litter, dead roots, rhizodeposition, microbial-derived), degree of microbial processing and decomposition, and turnover times (Lehmann and Kleber, 2015). SOC is the main component of soil organic matter and varies in spatial and temporal dynamics.*

Lines 47 – 49: Is this true for PyOC as well? What is the primary mechanism of preservation for pyrogenic organic matter if not some level of biochemical recalcitrance?
You are correct, the primary mechanism for protection of PyOC is presumably biochemical recalcitrance, and we indicate this in lines 77-78 (original manuscript). Here we refer to the main stabilization mechanisms of SOC as a whole, not of each fraction separately. There has always been a good debate on this subject, but I think that in the last 10 years there has been more evidence of the importance of the physico-chemical protection vs the biochemical recalcitrance alone. We have added at the end of the paragraph the sentence:

*However, the hierarchy between stabilization mechanisms varies with the pedoclimatic context, land use and SOC fraction.*

Line 74: In the MAOM fraction, what is the mechanism of preservation of PyOM in the MAOM fraction? Is it occlusion within microaggregates that are smaller than 60/53 µm? Or can PyOM form organo-mineral association? I think it this section it would be worth discussing this fraction in a little more depth.
Thanks for your question. Intuitively I would imagine that PyOC is able to be found inside microaggregates smaller than 53 µm and also to be adsorbed to mineral surfaces. There does not seem to be many studies performing size/density fractionation and determination of PyOC under field conditions, but most of them investigate biochar additions. Zimmerman and Mitra (2017) suggest that PyOC (added as biochar) may enhance SOC stabilization via sorption and physical protection inside aggregates.
We searched in the literature regarding the physico-chemical stabilization of PyOC and found the following reference:

Burgeon, V., Fouché, J., Leifeld, J., Chenu, C., and Cornélis, J.-T.: Organo-mineral associations largely contribute to the stabilization of century-old pyrogenic organic matter in cropland soils, Geoderma, 388, 114841, https://doi.org/10.1016/j.geoderma.2020.114841, 2021.

The authors found that PyOC is found in the free light fraction (analogous to free POM), and also occluded inside aggregates (macro and microaggregates) and sorbed onto mineral phases of the clay and silt fraction. We have rephrased the paragraph as follows:

*PyOC can be found in both POC and MAOC fractions (Lavallee et al., 2019). Beyond the biochemical recalcitrance, which would be the main mechanisms of PyOC in the POC fraction (e.g., free PyOC non-associated to clay and silt sized mineral particles), PyOC can also interact with mineral phases and be protected inside microaggregates or adsorption (Burgeon et al., 2021). Zimmerman and Mitra (2017) suggest that PyOC (e.g., naturally occurring or added as biochar) may promote the stabilization of non-PyOC by enhancing the creation of microaggregates and sorption onto existing organo-mineral complexes.*

Lines 77 – 78: This is kind of what I mean in my comment above, it seems contradictory.
Thanks for your comment. I don't think it is necessarily contradictory. I believe that the main SOC stabilization mechanism varies depending if we consider bulk SOC or a specific SOC fraction, and will also change on a case by case. Although globally, I think that physico-chemical stabilization is more important than biochemical recalcitrance. Also, at some point these mechanisms are working simultaneously. In the case of the PyOC fraction it is generally accepted that biochemical recalcitrance is the main protection mechanism, but that is not the case for MAOC.

Lines 88 – 89: It would be great to have a citation here for either how fractions can inform management, or how they can be incorporated into policy.
For example, the Australian National Soil Strategy has among its objectives to increase and maintain SOC, and this includes to develop cost-effective ways to estimate and model SOC. We hope that our maps could be used for that purpose in the future. In the meantime, we include a reference to Dangal et al. (2022) where they use maps of measurable SOC fractions as input for a biogeochemical model and project future SOC under future climate and land cover scenarios. Maybe it does not show how management is informed, but it is an application of similar maps in that direction.

Dangal, S. R. S., Schwalm, C., Cavigelli, M. A., Gollany, H. T., Jin, V. L., and Sanderman, J.: Improving Soil Carbon Estimates by Linking Conceptual Pools Against Measurable Carbon Fractions in the DAYCENT Model Version 4.5, Journal of Advances in Modeling Earth Systems, 14, e2021MS002622, https://doi.org/10.1029/2021MS002622, 2022.

Lines 139-142: Could you add a discussion either here or later in section 2.3 related to the pre-processing of the data from the different libraries? Were the data smoothed and corrected using Savitzky-Golay or the like, and how did that differ across the different libraries? If they are raw spectra, please specify that they were received in that form.
Thanks for your comment. We have indicated the pre-processing of the different spectral libraries but later in the section, before the piecewise direct standardization step.

*SCaRP spectra were baseline-corrected and mean-centered prior subsequent analyses but were not subject to additional pre-processing. Pre-processing of the AusSpecNIR spectra consisted of spectral trimming (453-2500 nm), a Savitsky-Golay smoothing filter, conversion of reflectance to absorbance, and standard normal variate transformation. AusSpecMIR and AusSpecMIR2 were both pre-processed with the following steps: 1) spectral resolution and range harmonisation. All spectra were resampled using a smoothing spline function to a common 2 $cm^{-1}$ resolution. The spectral range was set to 6500 $cm^{-1}$ to 598 $cm^{-1}$, 2) Savitsky-Golay smoothing filter with 22 $cm^{-1}$ local neighbourhood, 3) conversion from reflectance to absorbance units, and 4) standard normal variate transformation.*

Line 219-220: Are there data associated with the C recovery of the fractions that might explain some of the poor matching between TOC and fraction sums? Soluble and dissolved carbon can be lost throughout the fractionation process, which may bias predictions.
Thanks for this question. Baldock et al. (2013) reported that the TOC recovery from the laboratory SOC fractionation (SCaRP samples) ranged from 76 to 142%, with an average TOC recovery of 102% and a standard deviation of 7.4%, and 86% of the samples yielded a recovery of 90–110%. In this case, I think the mismatch comes mainly from the prediction error of the spectral models, although some propagation of error from the

SOC fractionation data occurs as well. For example, for AusSpecNIR, AusSpecMIR and AusSpecMIR2, the SOC fraction concentration was calculated from predictions of SOC distribution between the three fractions, and predicted TOC. We have added:

*The mismatch between the sum of SOC fractions and measured TOC is most likely derived from the prediction error of the spectral models, and in minor extent from the TOC recoveries of the laboratory SOC fractionation data.*

Line 235: Can you clarify the difference between ilr1 and ilr2 in this line? Looking through the equations and text I think I can piece it together, but it would be helpful to the reader to make it explicit.
Thank you for the suggestion, but in this case, I think that if the reader is interested in understanding the ilr transformation in depth, they can read the reference Egozcue et al. (2003). I think that the essential information is that this transformation allows to model compositional data and reduces the variables in one dimension (in these case, from three to two variables). But also, given the nature of the transformation it is hard to interpret ilr1 and ilr2. They can also write the equation for ilr1 and ilr2 by substituting the terms (i=1 and i=2) in equation 4 (equation 1 in the previous version), so that we don't add more equations to the manuscript.

Line 280: How much of the dataset was void-filled? Can you provide a percentage for total data interpolation across covariates?
We did not calculate the exact percentage but it would be minimal. I would say less than 0.5% of the study area. These voids corresponded to some waterbodies in the DEM mainly.

Line 387 and on: As my fellow reviewer noted, there are inconsistencies in the reporting of significant digits and errors in the results section of the manuscript. At the risk of being redundant, I recommend the authors carefully check the figures they present for consistency and utility.
Thanks for your comment. We will revise the number of significant digits throughout he manuscript.

Line 498-499: Are most agricultural lands in the Mediterranean biome irrigated? It could be worth highlighting the proportion irrigated either here or in the discussion.
Thank you for this question. We have calculated the proportion of irrigated agricultural land (pastures and cropping) for the Mediterranean biome using the land use map of 2018 (ABARES), and surprisingly we found that it is less than 2% of the area (1.6%). We have included this in the text:
*where around 1.6% of the agricultural area (pastures and cropping) was irrigated (Abares, 2022)*

Australian Bureau of Agricultural and Resource Economics and Sciences (ABARES). Land use: https://www.agriculture.gov.au/abares/aclump/land-use, last access: 20/09/2022.

Line 517: Higher mean sand content or higher mean SOC concentration?
I have corrected the manuscript. It is sand content.

Line 522-523: I think this sentence is confusing, I recommend rewording. Currently it reads as if the authors are discussing the total proportion of MAOC across Australia.
Thanks for your suggestion. We have reworded the sentence as "Conversely, the proportion of SOC found as MAOC is around 60% in Australian temperate grasslands, savannas and shrublands, 50 % in Mediterranean forests and woodlands, ~ 54-64% in temperate forests and 65-65 % in tropical (Table 3)."

Line 541: Please clarify the directionality of the relationship between POC and MAT.
Thanks for the suggestion. We have clarified that "The content of free POC decreased with an increase of MAT in topsoils (0-30 cm), whereas free POC increased slightly with MAT in subsoils, and occluded POC increased with MAT at all depths".

Lines 552 - 554: I'm glad you mention this here -- I was going to recommend something along these lines as a justification for not including these co-variates.
Thanks for your comment. I extracted the available data on exchangeable Ca and Mg, oxalate and dithionite extractable Fe and Al from the SoilDataFederator, but the spatial coverage was very irregular. Different regions had just few observations for each chemical property, which I could barely match with MAOC data. Definitely it would be great to examine these relationships in the future, as it has been done in other regions of the world.